# Biomimetic organo-hydrogels reveal the adipose tissue local mechanical anisotropy regulates ovarian cancer invasion

Jordi Gonzalez-Molina [1,2] ✉, Parisa Nabili[1,3], Daniele Marciano [2], Sara Abdelnabi[1], Okan Gultekin [1,3], Mohammed Fatih Rasul[3,4], Yikun Zhang[2], Clémence Nadal[2], Alexandra Chrysanthou[2], Twana Alkasalias [3,5], Sahar Salehi[3,6], Frances Balkwill [7], Kaisa Lehti [1,8] & Julien E. Gautrot [2] ✉

High-grade serous ovarian cancer, the most common and aggressive ovarian cancer subtype, frequently metastasises to visceral adipose tissues. In these tissues, the extracellular matrix through which ovarian cancer cells migrate is constrained by the presence and preponderance of adipocytes. How cells migrate in this unique environment is not known, yet critical to understanding metastatic progression. To study these processes, we developed biomimetic organo-hydrogels that recreate structural, mechanical, and biochemical properties of human adipose tissues. We show that ovarian cancer cells present invasive tropism towards organo-hydrogels, replicating the behaviour observed in native adipose tissues. This migration is facilitated by the mechanical anisotropy and microstructure of organo-hydrogels and adipose tissues, allowing the formation of migratory tracks. These results highlight the contribution of adipocytes to tissue biophysical features as a key regulatory factor of ovarian cancer cell migration and demonstrate that organo-hydrogels are particularly relevant tools to develop in vitro models of complex tissue architectures with high cellularity.

Ovarian cancer is the deadliest gynaecological malignancy[1]. High-grade serous ovarian cancer (HGSC) is the most common and aggressive subtype of ovarian cancer, accounting for approximately 70% of all cases[2]. Ovarian cancer commonly disseminates trans-coelomically by shedding cancer cells from the surface of the fallopian tube and/or ovary into the peritoneal cavity[3]. After detachment, cells are carried by the ascitic fluid, attach to the mesothelial cell layer, and invade the adipose tissue of the omentum and the peritoneum. This tropism for adipose tissues, which has also been observed in the alternative haematogenous route of metastasis, is not fully understood[4,5].

Cell migration within tissues is a fundamental process for the establishment and progression of metastases[6]. By using natural and engineered biomaterials, recent studies have revealed that the biochemical and biophysical properties of the extracellular matrix (ECM) control cell migration and can dictate the migratory mode adopted by cells[7,8]. These regulatory mechanisms are particularly relevant to understand cancer cell invasion into ECM-rich tissues such as bone, as

[1]Department of Microbiology, Tumor and Cell Biology, Karolinska Institutet, Solnavägen 9, Solna, Sweden. [2]School of Engineering and Materials Science, Queen Mary University of London, Mile End Road, London, United Kingdom. [3]Department of Women's and Children's Health, Division of Obstetrics and Gynecology, Karolinska Institutet, Solnavägen, Solna, Sweden. [4]Department of Pharmaceutical Basic Science, Faculty of Pharmacy, Tishk International University, Erbil, Iraq. [5]General Directorate of Scientific Research Centre, Salahaddin University-Erbil, Erbil, Iraq. [6]Department of Pelvic Cancer, Theme Cancer, Karolinska University Hospital, Stockholm, Sweden. [7]Barts Cancer Institute, Queen Mary University of London, Charterhouse Square, EC1M6BQ, London, UK. [8]Department of Biomedical Laboratory Science, Norwegian University of Science and Technology - NTNU, Erling Skjalgssons gate 1, Trondheim 7491, Norway. ✉e-mail: jordigonzalezmolina@gmail.com; j.gautrot@qmul.ac.uk

well as the stroma of desmoplastic tumours and some sarcomas[9–11]. Conversely, adipose, muscle, and epithelial tissues display a high cellularity[10,12,13]. Adipose tissues are largely composed of adipocytes, while the volume occupied by the ECM-containing interstitial space is minimal. However, upon ovarian cancer invasion, adipocytes shrink and are progressively replaced by an ECM-rich stroma[14]. The processes regulating ovarian cancer cell invasion into adipose tissues and the impact of the unique microstructural and biophysical properties of the adipose microenvironment remain ill-defined.

In this work, we develop an organo-hydrogel (OHG) model that captures the microstructure, mechanics and ECM biochemistry of adipose tissues. Using this model, we replicate the tropism of ovarian cancer cells observed in native adipose tissues and demonstrate that the anisotropic mechanical properties and microstructure of adipose tissues play a significant role in this phenomenon. We uncover that ovarian cancer cell invasion is independent of matrix degradation but requires contractile cell-mediated forces to generate migratory tracks at the adipocyte-ECM interface. We demonstrate that the size of adipocyte-mimicking microdroplets and their density regulate the mode of cell migration and effective tissue invasion, capturing the close relationship linking the development of metastases and tissue microstructural changes. This study uncovers the biophysical regulation of cancer cell migration in adipose tissues and presents an in vitro platform to investigate and model adipose tissue metastasis.

## Results

### Organo-hydrogels mimic human peritoneal adipose tissues
To recreate the microstructure of visceral adipose tissues, we designed organo-hydrogels (OHGs) comprising microdroplets mimicking the adipocyte phase, separated by an aqueous ECM-rich phase (Fig. 1a). For comparison, we used human peritoneal tissues, obtained from HGSC debulking surgeries, devoid of or with minimal presence of cancer cells ($0.38 \pm 0.21\%$ PAX8 + nuclei of total tissue nuclei (s.e.m.); $0.75 \pm 0.44$ PAX8 + nuclei mm$^{-2}$ tissue (s.e.m.)) (Supplementary Fig. 1a). The peritoneal adipocyte-like emulsion was formulated using ~70 μm diameter oil microdroplets (silicone oil for inertness) stabilised by bovine serum albumin (BSA; Fig. 1a–c). The microdroplet size was comparable to the average size of peritoneal adipocytes, though inter-patient average adipocyte diameter ranged between 49 and 97 μm (Fig. 1c; patient age ranged between 48–78 years, and body mass index ranged between 19–36). To gel these microdroplets into OHGs, we selected collagen type I as the ECM phase because it is the main ECM component of adipose tissues (Supplementary Fig. 1b)[10,14]. After gel formation, the volume fraction occupied by oil was $90 \pm 1\%$ at the OHG core and $91 \pm 2\%$ (s.d.) in peritoneal adipose tissues (Fig. 1d). We compared the bulk mechanical properties of human peritoneal tissues, including with (i.e., adipose tissue) and without (i.e., connective tissue) visible fat, with those of adipose tissue-mimicking OHGs and collagen matrices (with identical collagen concentration than in the aqueous phase of OHGs). The storage moduli of OHGs and collagen gels were found to be comparable to peritoneal connective tissues and lower but within the range of peritoneal adipose tissues (Fig. 1e). Similarly, the stress-relaxation profile of OHGs closely matched those of peritoneal tissues (Fig. 1f). Analysis of local mechanical properties by force probe microscopy revealed marginally higher Young's moduli of OHGs ($1300 \pm 41$ Pa (s.e.m.); range: 210–2600) compared to peritoneal adipose tissues ($1100 \pm 74$ Pa (s.e.m.); range: 90–7900; $p = 0.0473$; Supplementary Fig. 1c). Collectively, these data indicate that the structural and mechanical properties of OHGs resemble those of native peritoneal adipose tissues.

### Adipose invasion tropism is recapitulated in organo-hydrogels
To evaluate the invasive behaviour of ovarian cancer cells, we embedded pre-formed spheroids of 7 human ovarian cancer cell lines in OHGs and evaluated changes in spheroid area as a function of time.

This model mimics the invasion occurring after initial colonisation of the tissue. After 7 days, the spheroid area increased in Tyk-nu, Tyk-nu.CPR, OVCAR8 and CAOV3 but not in OVCAR3, OVCAR4, and Kuramochi (Fig. 1g). To evaluate the impact of OHGs on spheroid expansion, in comparison with fibrous collagen matrix, we compared spheroid area changes in cells embedded in collagen gels and OHGs. OVCAR8 and CAOV3 presented increased spheroid area, broke spherical symmetry and showed invasiveness into OHGs, while OVCAR3, Kuramochi, and OVCAR4 were non-invasive (Fig. 1h and Supplementary Fig. 2). Tyk-nu and Tyk-nu.CPR spheroids broke spherical symmetry and presented invasiveness in both collagen and OHGs. Collagen matrix pore size and density can alter 3D cell migration[8]. Collagen gels in bulk and in the ECM phase of OHGs displayed equivalent collagen density and average pore size, indicating that the presence of the oil microdroplets in OHGs did not alter the collagen phase microstructure, but significantly affected the spheroid area and invasive phenotype of corresponding cells (Supplementary Fig. 3). To determine whether OHGs altered proliferation or apoptosis, we compared Ki67 and cleaved-caspase 3 expression in the distinct cell lines, respectively. Cleaved-caspase 3 expression was generally low, but Ki67 expression was enhanced in OHGs in both CAOV3 and OVCAR8, especially at the periphery of the spheroid (Supplementary Figs. 2, 4). These results are in line with other reports showing that pro-invasive matrices promote cell proliferation in cancer spheroids[15].

Initial ovarian cancer cell invasion of the adipose tissue occurs after attachment of cells present in the ascites fluid. To mimic this process, we established an organotypic invasion assay where cells were seeded on top of cell-free collagen gels or OHGs. In agreement with the spheroid model, OVCAR8 showed greater depth of invasion in OHGs than in collagen gels, where invasion was minimal (Fig. 1i, j). Next, we investigated the invasive behaviour of OVCAR8 cells ex vivo into human tissues by seeding mRFP-OVCAR8 cells on top of adipocyte- or connective tissue-rich regions of the peritoneum. Recapitulating the behaviour in the hydrogel models, OVCAR8 depth of invasion was greater in adipose tissues than in connective tissues, where cells largely remained at the surface (Fig. 1k, l). In contrast, collagen gel-invading Tyk-nu cells were able to invade into the connective tissue (Supplementary Fig. 5a, b). Subsequently, we directly compared cell invasion into peritoneal adipose tissue and OHGs, observing a similar invasion response (Fig. 1m, n). Next, we investigated the behaviour of HGSC patient-derived malignant ascites cells. Malignant ascites cells often form heterogeneous cell aggregates of cancer (PAX8 + ) and stromal/mesothelial cells (PAX8-) (Supplementary Fig. 5c)[3]. The organotypic invasion of both PAX8- and PAX8+ ascites cells was enhanced in OHGs compared to collagen gels (Supplementary Fig. 5d, e). Moreover, the embedding of malignant cell aggregates indicated that cell invasion of CK7 + cells (expressed in cancer cells and mesothelial cells) and cell proliferation of both PAX8- and PAX8 + cells was enhanced in OHGs compared to collagen gels (Supplementary Fig. 5f, g). Altogether, these data indicate that the tropism shown by ovarian cancer cells towards ex vivo adipose tissues is recapitulated in OHGs without the presence of chemotactic gradients.

### ECM adhesion and TGFβ regulate ovarian cancer invasion
Cancer cells invade into 3D microenvironments through single-cell or collective modes of migration, based on both ECM adhesion-dependent and -independent mechanisms[7]. To investigate the impact of ECM adhesion and signalling on ovarian cancer cell invasion, we embedded individual OVCAR8 cells, to minimise the effect of cell-cell adhesion, in collagen or OHGs for 48 h and investigated the phosphorylation of focal adhesion kinase (pFAK) and myosin light chain (pMLC). The expression of both phosphorylated proteins was found to be enhanced in OHGs (Fig. 2a, b). Concurrently, nuclear localisation of yes-associated protein (YAP) was higher in OHGs. Although the surface of the oil microdroplets of OHGs did not present

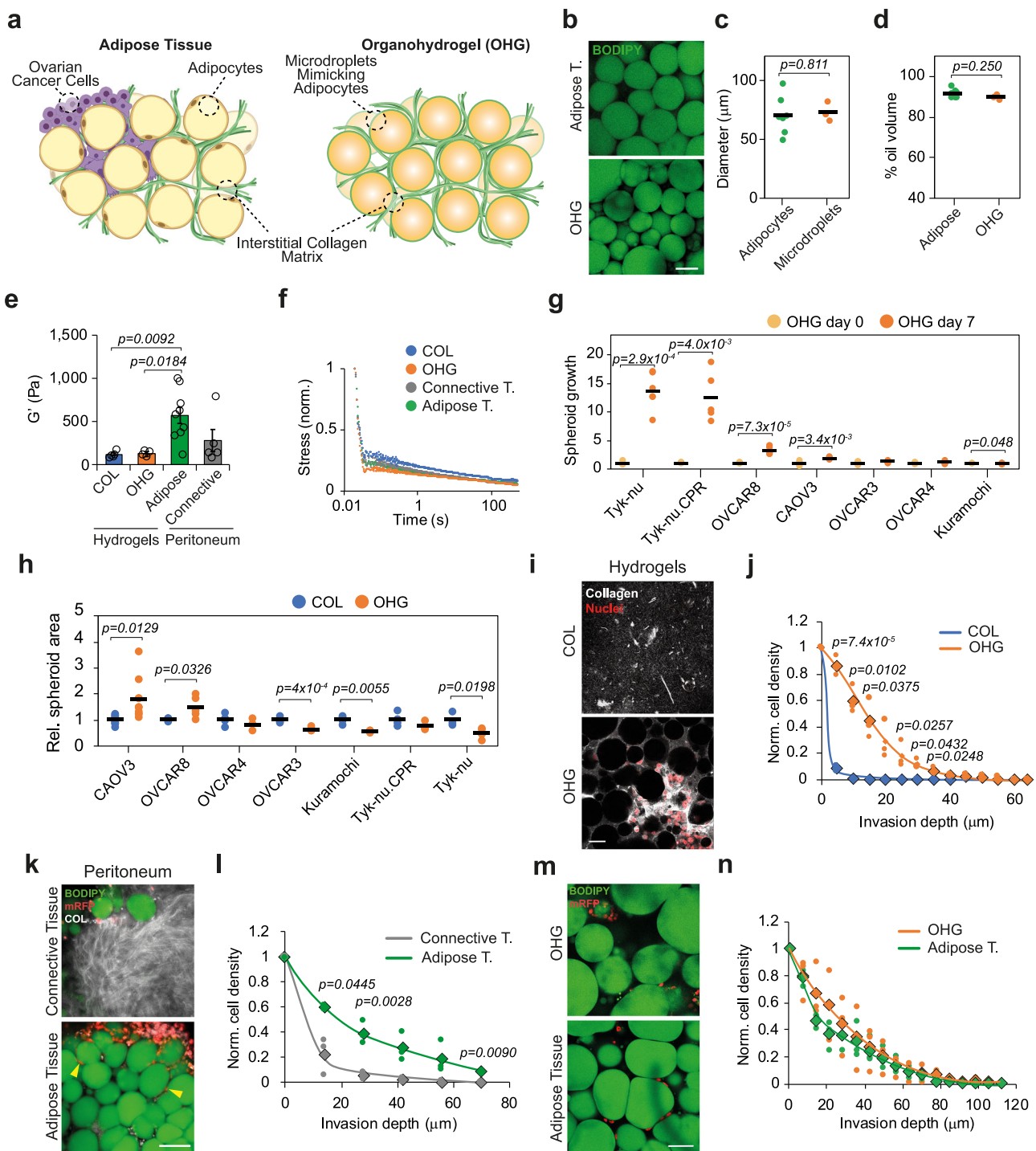

adhesion ligands, we observed that cells spread at the microdroplet-collagen interface (Fig. 2c). This led to alterations in nuclear morphology, such as increased aspect ratio and reduced circularity in OHGs (Fig. 2d). Similarly, we observed a greater nuclear aspect ratio and lower circularity in cells invading human peritoneal adipose tissue compared to cells at the tissue surface (Supplementary Fig. 6a, b). These observations suggest that the microstructure of adipose tissues and OHGs dictates cell confinement and spreading, and impose physical barriers to cell migration. To evaluate the impact of the altered ECM adhesion and signalling on invasion, we treated embedded OVCAR8 spheroids with integrin β1 blocking antibody or with FAK, myosin-II, Rho-associated protein kinase (ROCK), or YAP inhibitors. All inhibitions except that of YAP reduced or blocked cell invasion into

OHGs (Fig. 2e and Supplementary Fig. 6c), demonstrating that adhesion to the ECM and associated downstream activation of the acto-myosin machinery is required for invasion into OHGs. Similarly, inhibition of FAK, which showed the strongest inhibition in OHGs, resulted in decreased invasion into human peritoneal adipose tissue (Fig. 2f, g). In addition, in HGSC ascites cells embedded in OHGs, inhibition of FAK diminished the proliferation of PAX8- and PAX8 + cells, although the response of PAX8 + cells in distinct donors was heterogeneous (Supplementary Fig. 6d). Finally, as confinement and nuclear deformations can cause DNA damage, in turn regulating migratory phenotypes[16,17], we compared the expression of the DNA damage marker γH2AX in cells after culture within collagen gels and OHGs. Expression of γH2AX was found to be low and comparable within both materials

**Fig. 1 | Ovarian cancer invasive behaviour is reproduced in adipose tissue-mimicking OHGs. a** Schematic representation of adipose tissue-mimicking collagen-based organo-hydrogels (OHGs). **b** BODIPY (green) staining of human peritoneal adipose tissue and OHG (representative images from $n = 7$ patients and $n = 3$ OHGs). **c** Quantification of human peritoneal adipocyte ($n = 100$ adipocytes/tissue from 7 patients) and silicone oil microdroplet ($n = 100$ microdroplets from 3 OHGs) diameter. **d** Quantification of volume fraction occupied by oil in peritoneal adipose tissues ($n = 7$ patients) and OHGs ($n = 3$ gels). **e** Storage moduli of hydrogels and human peritoneal tissues ($n = 5$ gels; $n = 9$ adipose and $n = 5$ connective tissues). **f** Normalised stress-relaxation curves of hydrogels and human peritoneal tissues ($n = 5$ gels or tissues). **g** Spheroid area after 7 d relative to day 0 of multiple ovarian cancer cell lines ($n = 3$ spheroids). **h** Spheroid area comparison of ovarian cancer cell lines after 7 d in collagen or OHGs relative to the collagen average ($n = 3$ spheroids). **i** Collagen-I (grey) and mRFP-OVCAR8 nuclei (red) staining of OVCAR8 cells seeded on top of collagen or OHG at 25 μm gel depth after 7 d in culture (representative images from $n = 3$ experiments). **j** Quantification of hydrogel organotypic invasion of OVCAR8 cells after 7 d ($n = 3$ experiments). Rhombuses

indicate the average. **k** BODIPY (green), mRFP-OVCAR8 (red), and collagen I (grey) staining of peritoneal tissue explants and mRFP-OVCAR8 cells after 7 d in culture (representative images from $n = 3$ experiments). Arrowheads indicate invading mRFP-OVCAR8 cells. **l** Quantification of mRFP-OVCAR8 organotypic invasion into human peritoneal tissues after 7 d ($n = 3$ tissues from distinct donors). Rhombuses indicate the average. **m** BODIPY (green) and mRFP (red) staining of mRFP-OVCAR8 cells in peritoneal tissue explants and OHGs (42 μm depth) after 7 d culture (representative images from $n = 3$ experiments). **n** Quantification of mRFP-OVCAR8 organotypic invasion into human peritoneal tissues or OHGs after 7 d ($n = 3$ tissues from distinct donors or gels). Rhombuses indicate the average. For the data in (**c**, **d**, **g**, **h**, **j**, **l** and **n**), a two-sided unpaired $t$ test was performed. One-way analysis of variance (ANOVA) with Tukey's correction for multiple comparisons was performed for the data in (**e**). Error bars in (**e**) represent the s.e.m. Scale bars, 50 μm (**b**, **i**, **m**), 100 μm (**k**). Components of (**a**) have been created in BioRender. Gautrot, J. (2025) https://BioRender.com/rk3jchz. Source data are provided as a Source Data file.

(Supplementary Fig. 6e, f). Thus, these data indicate that ECM-mediated adhesion, cytoskeletal contractility and downstream signalling are fundamental for cell invasion into OHGs and native adipose tissues, but that associated confinement is not conducive of DNA damage, as in other reported microstructured environments[16,17].

To further understand the regulation of ovarian cancer invasiveness, we conducted a gene set enrichment analysis of differentially expressed genes between OHG-invading and non-invading cell lines (Fig. 1g and Supplementary Fig. 7a, b). The most enriched gene sets in the downregulated genes were bone marrow stromal, cell morphogenesis, and cell-cell junction. Loss of cell-cell junctions typically occurs during epithelial-to-mesenchymal transition (EMT)[18]. A key inducer of EMT is the transforming growth factor-β (TGFβ), which has been associated with ovarian cancer metastasis and omental invasion[18–21]. Thus, we investigated the impact of TGFβ signalling on OHG invasion. Inhibition of TGFβ signalling in OVCAR8 reduced spheroid invasion while treatment with TGFβ enhanced CAOV3 invasion in OHGs (Fig. 2h–k). Similarly, inhibition of TGFβ reduced CK7 + spheroid size and the number of invading CK7 + single HGSC malignant ascites cells embedded in OHGs (Supplementary Fig. 7c–e). Moreover, inhibition of TGFβ signalling diminished organotypic invasion of PAX8- and PAX8 + ascites cells (Supplementary Fig. 7f, g). Finally, TGFβ treatment induced OHG invasiveness in the initially non-invading Kuramochi cell line (Fig. 2l, m). These data indicate that TGFβ signalling is pivotal to the regulation of ovarian cancer cell invasion into adipose tissue-mimicking OHGs.

To better understand the role of the ECM phase of OHGs on cell invasion, we explored the use of norbornene-functionalised hyaluronic acid (NB-HA) as the ECM phase instead of collagen. This system allowed us to investigate whether the observed cell behaviours are collagen-dependent and provided a higher degree of control over ECM features such as adhesion ligand presentation, matrix degradability and mechanics. We formed OHGs by photo-crosslinking microdroplets with NB-HA and di-cysteine matrix metalloproteinase (MMP)-cleavable peptides (Fig. 3a). The storage modulus of NB-HA OHGs was comparable to that of collagen-based OHG and within the range of peritoneal adipose tissues (Fig. 3b). Stress relaxation in NB-HA OHGs was low compared to the fast-relaxing collagen-based OHGs (Fig. 3c). Despite this difference, OVCAR8 cells invaded NB-HA OHGs presenting the collagen-derived adhesion motif GFOGER[22] and similarly presented cell spreading at the matrix-microdroplet interface, albeit to a reduced level compared to their migration in collagen-based OHGs (Fig. 3d–g). During the development of omental metastases, the composition of the ECM evolves, with increasing presence of non-collagenous proteins such as fibronectin and osteopontin, which contain the cell adhesion motif RGD[14]. Thus, we compared OVCAR8 invasion into OHGs containing

GFOGER or RGD. Invasion into OHGs was comparable in both types of OHGs (Fig. 3f–h). These results indicate that, although OVCAR8 invasion into OHGs is integrin-mediated, it is not restricted to collagenous ECM phases.

## Invasion is regulated by the mechanical anisotropy of adipose tissues

Invasion through dense matrices often requires the proteolytic degradation of the ECM[7]. However, as cell migration in OHGs was found to be associated with spreading at microdroplet-ECM and adipocyte-ECM interfaces (Fig. 2c and Supplementary Fig. 6a), we hypothesised that the physico-chemistry of these interfaces, but not MMP activity, is critical to the migratory phenotype observed. First, we investigated the importance of ECM degradation on invasion into adipose tissues. The invasion of OVCAR8 spheroids embedded in collagen-based OHGs was not altered upon treatment with a pan-MMP inhibitor (Fig. 3i, j). Likewise, pan-MMP inhibition did not alter OVCAR8 invasion into human peritoneal adipose tissue explants or the organotypic invasion and the proportion of proliferating malignant ascites cells in OHGs (Fig. 3k, l and Supplementary Fig. 8a–c). To further confirm that MMP activity and enzymatic degradation are not required for invasion, we embedded spheroids in NB-HA-based OHGs and compared the invasion of cells in systems crosslinked with an MMP-cleavable peptide or non-degradable polyethylene glycol-dithiol (PEGDT). Cell invasion and spheroid area were comparable in both MMP-cleavable and PEGDT-crosslinked OHGs, where cells migrated in direct contact with microdroplets (Fig. 3m and Supplementary Fig. 8d).

These data indicate that invasion in adipose tissues and OHGs is MMP-independent and could be enabled by migration at the adipocyte-ECM interface. This contrasts with cell migration through dense collagen matrices, typically dependent on local degradation[23,24]. In agreement with these observations, individual cells embedded in OHGs spread at the microdroplet-ECM interface, but not in equally dense collagen matrices, upon pan-MMP inhibition (Fig. 4a). Notably, the microdroplets and peritoneal adipocytes in contact with OVCAR8 cells presented deformed interfaces at contact points (Fig. 4b and Supplementary Fig. 9a). This further suggests that cell migration at the adipocyte-ECM interface is adhesion-mediated, resulting in forces that can deform adipocyte/microdroplet interfaces. Indeed, blocking cell contractility by inhibiting ROCK abolished the invasion into OHGs and adipose tissue explants, reducing nuclear deformation at the cell invasive front (Fig. 4c and Supplementary Fig. 9b–d). Similarly, ROCK inhibition diminished organotypic OHG invasion of HGSC patient ascites cells (Supplementary Fig. 9e, f).

Subsequently, we investigated whether microdroplet deformation and the associated mechanical anisotropy of microdroplet-

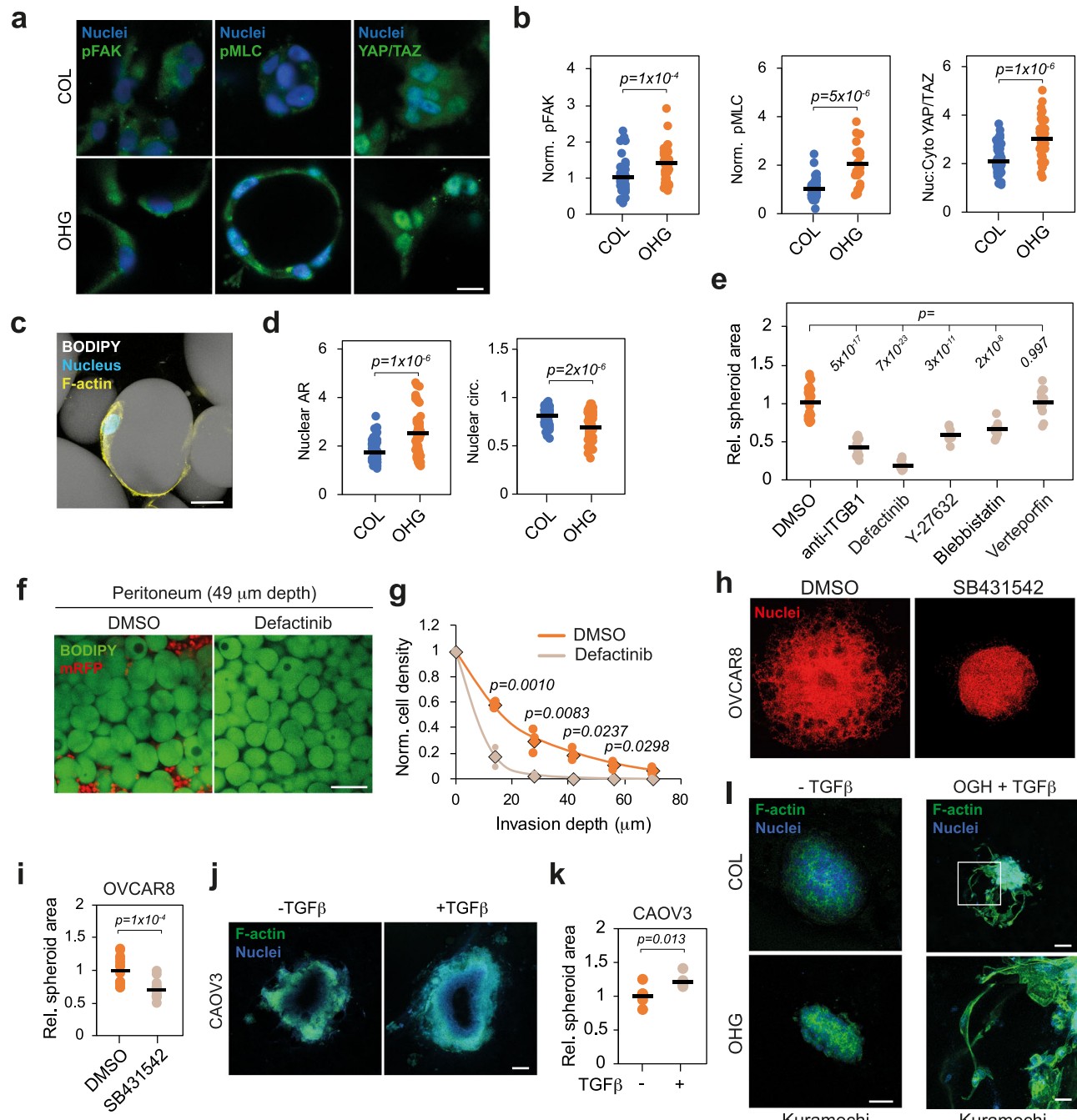

**Fig. 2 | ECM adhesion and TGFβ signalling regulate cell invasion in OHGs.**
**a** pFAK, pMLC, YAP/TAZ (green; left to right) and nuclei (blue) staining of OVCAR8 cells in collagen or organo-hydrogel (OHG) after 48 h in culture (representative images from *n* = 3 gels). **b** Quantification of immunofluorescence signal intensity of cytoplasmic pFAK, pMLC, relative to the average in collagen and cytoplasmic:nuclear ratio of YAP/TAZ in OVCAR8 cells embedded in collagen or OHG for 48 h (*n* = 24 cells). **c** F-actin (yellow), nucleus (blue) and BODIPY (grey) staining of an OVCAR8 cell spread at the ECM-microdroplet interface. **d** Shape descriptors of OVCAR8 nuclei after 24 h in collagen or OHG (*n* = 53 nuclei). **e** Relative spheroid area of OVCAR8 after 7 d in OHG (*n* = 10 spheroids). **f** BODIPY (green) and mRFP (red) staining of peritoneal adipose tissue explants and mRFP-OVCAR8 cells (49 μm into the tissue from the surface) after 7 d in culture (representative images from *n* = 3 experiments). **g** Quantification of mRFP-OVCAR8 organotypic invasion into

human peritoneal tissues after 7 d (*n* = 3 tissues from distinct donors). Rhombuses indicate the average. **h** Nuclei (red) of OVCAR8 cells in OHGs after 7 d treatment (representative images from *n* = 14 spheroids). **i** Quantification of spheroid area relative to DMSO control of OVCAR8 cells in OHGs (*n* = 14 spheroids). **i** F-actin (green) and nuclei staining of CAOV3 spheroids embedded in OHGs for 7 d (representative images from *n* = 6 spheroids). **k** Quantification of spheroid area in CAOV3 cells in OHGs after 7 d (*n* = 6 spheroids). **l** F-actin (green) and nuclei (blue) staining of Kuramochi spheroids in collagen or OHG for 7 d with or without TGFβ (representative images from *n* = 3 spheroids). For the data in (**b**, **d**, **g**, **i** and **k**), a two-sided unpaired *t* test was performed. For data in (**e**), a one-way analysis of variance (ANOVA) with Tukey's correction for multiple comparisons was performed. Scale bars, 25 μm (**a**, **c**), 100 μm (**f**, **l**), 200 μm (**h**, **j**). Source data are provided as a Source Data file.

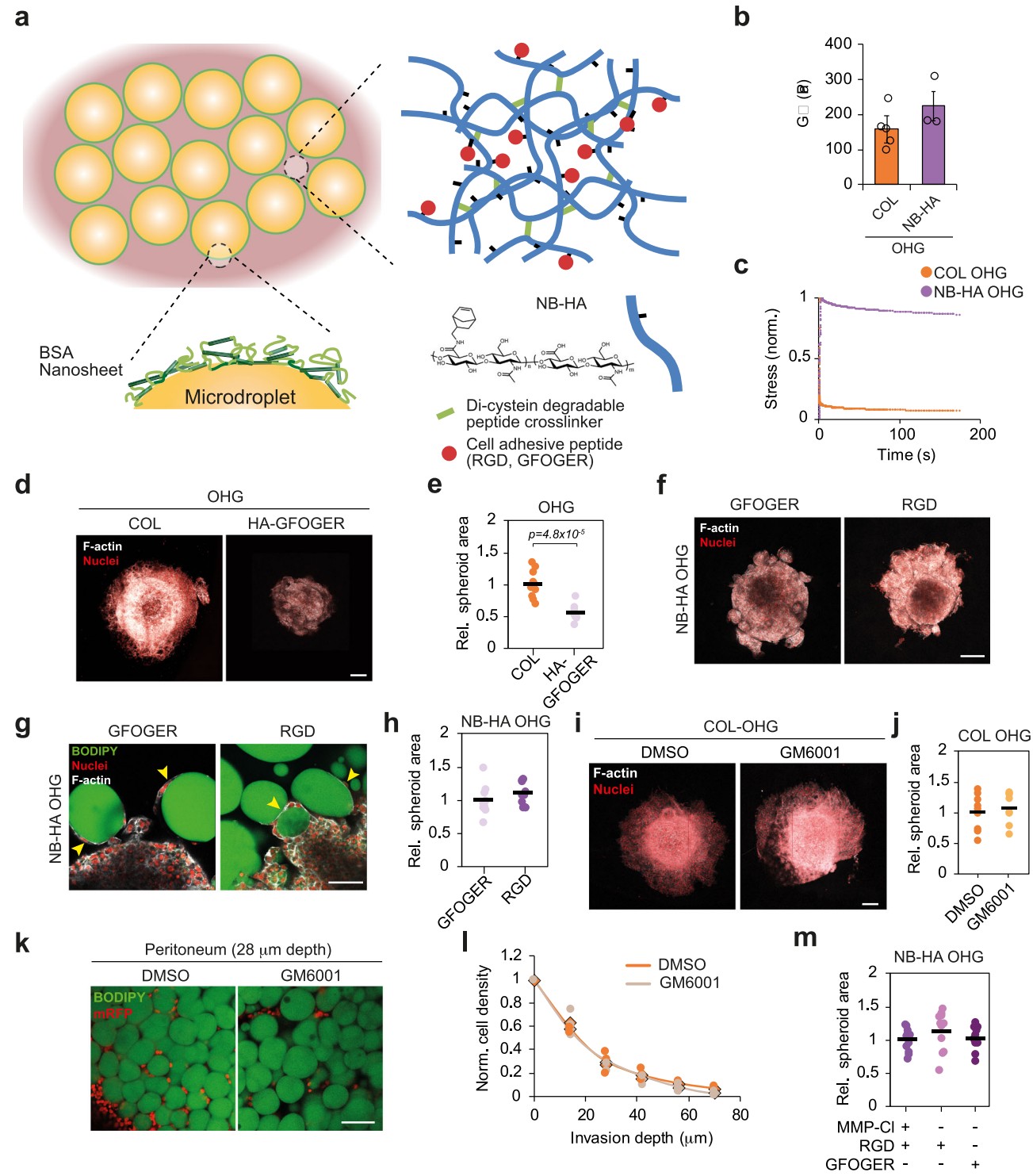

matrix interfaces facilitate cell invasion. Liquid PDMS microdroplets forming OHGs were replaced by elastomer microbeads based on crosslinked Sylgard 184 PDMS cured with increasing crosslinker concentrations (2–10 wt%) (Fig. 4d). The mechanics of liquid-liquid interfaces is highly anisotropic[25]. Oil microdroplets display extremely low interfacial shear moduli in the absence of tensioactive stabilisers, but display comparatively high indentation moduli. This can be reversed upon the adsorption of protein assemblies such as albumin nanosheets, raising interfacial shear moduli above 60 mN m$^{-1}$ (from 10$^{-5}$-10$^{-4}$ N m$^{-1}$ for bare interfaces), whilst indentation moduli are reduced 1.69-fold[25]. The introduction of crosslinked PDMS to form OHGs therefore aimed to reverse the mechanical

anisotropy at organic-aqueous interfaces within these materials, with Sylgard 184 bead formulations with storage shear moduli of 20, 147 and 260 kPa (corresponding to 2, 5 and 10 wt% crosslinking, respectively; Supplementary Fig. 10a, b). Circularity of elastomeric PDMS microbeads in contact with OVCAR8 cells was lower in soft compared to stiff PDMS microbeads (Supplementary Fig. 10c, d). Concurrently, invasion of OVCAR8 cells was significantly reduced in OHGs with stiff PDMS microbeads (Fig. 4e, f). Therefore, these results indicate that cell-mediated deformation of microdroplets and adipocytes, enabled by the mechanical anisotropy of associated interfaces, facilitates cell invasion in adipose tissues and adipose-mimicking OHGs.

**Fig. 3 | Cell invasion into adipose tissues is MMP-independent. a** Schematic representation of norbornene-functionalised hyaluronic acid (NB-HA)-based organohydrogels (OHGs) presenting matrix metalloproteinase (MMP)-degradable crosslinking peptides and cell adhesion ligands. **b** Storage moduli of collagen- and NB-HA-based OHGs (n = 5 collagen-based and n = 3 NB-HA-based OHGs; average ± s.e.m.). **c** Normalised stress-relaxation curves of collagen- and HA-NB OHG (n = 5 collagen-based and n = 3 NB-HA-based OHGs). **d** F-actin (grey) and nuclei (red) in OVCAR8 cells embedded in collagen- or NB-HA-based OHGs for 7 d (representative images from n = 9 spheroids). **e** OVCAR8 spheroid area in collagen- or NB-HA-based OHGs relative to collagen-based OHG average (n = 9 spheroids). **f** F-actin (grey) and nuclei (red) of OVCAR8 cells embedded in GFOGER- or RGD-presenting NB-HA OHG (representative images from n = 9 spheroids). **g** BODIPY (green), nuclei (red), and F-actin (grey) of OVCAR8 in GFOGER- or RGD-presenting NB-HA OHG (representative images from n = 9 spheroids). Arrowheads indicate cells in direct contact with oil microdroplets. **h** OVCAR8 spheroid area in GFOGER- or RGD-presenting NB-HA OHG after 7 d culture relative to GFOGER OHG average (n = 9 spheroids). **i** F-actin (grey) and nuclei (red) in OVCAR8 cells embedded in collagen-based OHGs for 7 d (representative images from n = 8 spheroids). **j** OVCAR8 spheroid area in collagen-based OHGs after 7 d culture relative to DMSO control (n = 8 spheroids). **k** BODIPY (green) and mRFP (red) staining of peritoneal adipose tissue explants and mRFP-OVCAR8 cells. **l** Quantification of mRFP-OVCAR8 organotypic invasion depth into human peritoneal tissues after 7 d (n = 3 tissues from distinct donors). Rhombuses indicate the average. **m** OVCAR8 spheroid area quantifications in MMP-cleavable or non-cleavable NB-HA OHG after 7 d culture relative to MMP-cleavable control (n = 11 spheroids). For the data in (**b**, **e**, **h**, **j** and **l**), a two-sided unpaired t test was performed. One-way analysis of variance (ANOVA) with Tukey's correction for multiple comparisons was performed for the data in (**m**). Error bars in (**b**) represent the s.e.m. Scale bars, 100 µm (**g**, **k**), 200 µm (**d**, **f**, **i**). Source data are provided as a Source Data file.

To investigate how cell adhesion at the microdroplet-ECM interface regulates invasion, we modified the NB-HA OHG system to include norbornene-BSA (NB-BSA) at the microdroplet surface, allowing functionalization of the microdroplet surface with cell adhesion peptides (Fig. 4g). Both OHGs generated with BSA and NB-BSA microdroplets, with NB-HA in the aqueous/gel phase, had comparable stiffness, stress-relaxation profiles and OVCAR8 spheroid invasive response (Supplementary Fig. 11a–d). We introduced RGD ligands either in the gel (NB-HA) phase and at the microdroplet surface of OHGs, or restricted to the microdroplet surface (Fig. 4h and Supplementary Fig. 11e). Although cell invasion into microdroplet-restricted RGD OHGs was significantly reduced compared to global-RGD OHGs, cells were still able to spread and migrate through the corresponding networks of microdroplets (Fig. 4h). Next, we investigated whether the mechanics of the microdroplet interface, which was found to modulate cell adhesion at liquid-liquid interfaces[26,27], regulates cell invasion in OHGs. We softened the stabilising protein assemblies by introducing β-casein (resulting in interfaces displaying interfacial shear storage moduli of $11.2 ± 1.0$ mN m$^{-1}$) and stiffened them by crosslinking with sulfosuccinimidyl 4-(N-maleimidomethyl)cyclohexane-1-carboxylate (sSMCC; $65.6 ± 13.7$ mN m$^{-1}$), compared to BSA-only controls ($35.7 ± 4.1$ mN m$^{-1}$) (Fig. 4i and Supplementary Fig. 12a, b)[28]. β-casein acts as a competitor for BSA adsorption, preventing the formation of the extensive protein nanosheet network that normally leads to the formation of interfaces with high interfacial shear moduli[29]. In contrast, the introduction of cyclohexyl residues via SMCC coupling enhances crosslinking of albumin nanosheets, increasing their interfacial shear moduli[28]. When adhesive motifs were only localised at the surface of microdroplets, cell invasion into OHGs with stiff interfaces was enhanced compared to softer interfaces (Fig. 4j, k). However, altering the interfacial mechanics of microdroplets forming OHGs presenting adhesion ligands in the NB-HA gel phase did not affect cell invasion (Supplementary Fig. 12c, d). Altogether, these data indicate that the localisation of cell adhesion motifs, together with the nanoscale mechanical properties of microdroplet interfaces, modulate cell invasion into adipose tissue-mimicking OHGs.

## Microdroplet and adipocyte size regulate cell invasion

In omental metastases, as the stromal compartment and ECM deposition increases, adipocyte diameter is reduced from approximately 70 to 25 µm[14]. Concurrently, the total volume occupied by adipocytes in the tissue diminishes progressively. Thus, we set out to determine whether microdroplet size and volume fraction in OHGs have an impact on cell invasion. We generated collagen-based OHGs with microdroplets of average diameters of ~140, 70, 25, and 15 µm, and 80, 60, 30, and 15% oil volume fractions, in which OVCAR8 spheroids were embedded (Fig. 5a and Supplementary Fig. 13a). At 80% oleophilic volume fraction, the storage modulus of OHGs increased exponentially with reduced microdroplet size

(Fig. 5b). The microdroplet size also affected the viscoelastic profile of OHGs, with slower stress-relaxation observed for OHGs formed with smaller microdroplets (Fig. 5c). At 80% oil volume fraction, the OVCAR8 spheroid size and invasion was maximal in 70 and 25 µm microdroplet OHGs, while both larger and smaller microdroplets presented reduced invasion (Fig. 5d, e). Diminishing the total oil volume fraction decreased the spheroid area in 140 and 70 µm microdroplet OHGs. However, spheroid areas in 25 and 15 µm microdroplet OHGs was maximal at 60 and 30% oil volume fractions, before decreasing again at 15% volume fraction. Overall, maximum invasiveness was achieved in biomimetic metastatic omentum OHGs, featuring 25 µm microdroplets, at 60 and 30% oil volume fractions[14].

Consistent with the results obtained with 70 µm microdroplet OHGs (Fig. 3i–l), inhibition of MMP activity in 25 µm microdroplet OHGs with varying oil volume fraction did not affect invasiveness (Supplementary Fig. 13b). In addition, in 25 µm microdroplet OHGs with increasing ECM fraction, the bulk storage modulus decreased and stress-relaxation increased (Supplementary Fig. 13c, d). These results suggest that adipocyte size can regulate cell invasion, even in the absence of the tissue stiffening normally observed in later disease stages[14,15]. Thus, we further compared OVCAR8 invasion into peritoneal tissues with high adipocyte volume fraction (89–95%), presenting different average adipocyte size and tissue stiffness (i.e., tissues from different patients). While cell invasion did not correlate with tissue stiffness (Supplementary Fig. 13e), a strong positive correlation was observed with increasing adipocyte size (Fig. 5f, g), indicating that at low ECM volume fraction, smaller adipocytes are a barrier for invasion, as observed in 25 and 15 µm microdroplet OHGs. These data indicate that microdroplet size and volume fraction, and therefore the microstructure of adipose tissues, are key regulators of cancer cell migration.

Next, we examined the mechanisms via which cancer cells sense microstructural features of the adipose microenvironment and regulate their invasion. Analysis of the invasive front in OHGs showed that cells migrate collectively in 70 µm but present single cell migration in 25 µm OHGs (Fig. 5d). A similar behaviour was observed in cells invading adipose peritoneal tissues with large and small adipocytes presenting collective and single cell migration, respectively (Supplementary Fig. 14a). Moreover, in 25 µm OHGs, nuclear aspect ratios increased and circularity decreased compared to 70 µm OHGs, suggesting that cells experience greater confinement in these conditions (Supplementary Fig. 14b, c). Indeed, the inter-microdroplet space is larger in 70 µm compared to 25 µm OHGs (Fig. 5h). Thus, the physical barriers imposed by 25 µm microdroplet OHGs are more prominent. However, the proportion of cells in contact with 25 µm microdroplets is higher (Fig. 5i), implying that these cells have additional migration paths that facilitate cell invasion. This may explain the enhanced invasion after partial reduction of the total oil volume fraction. This aligns with prior findings showing that enhanced migration paths

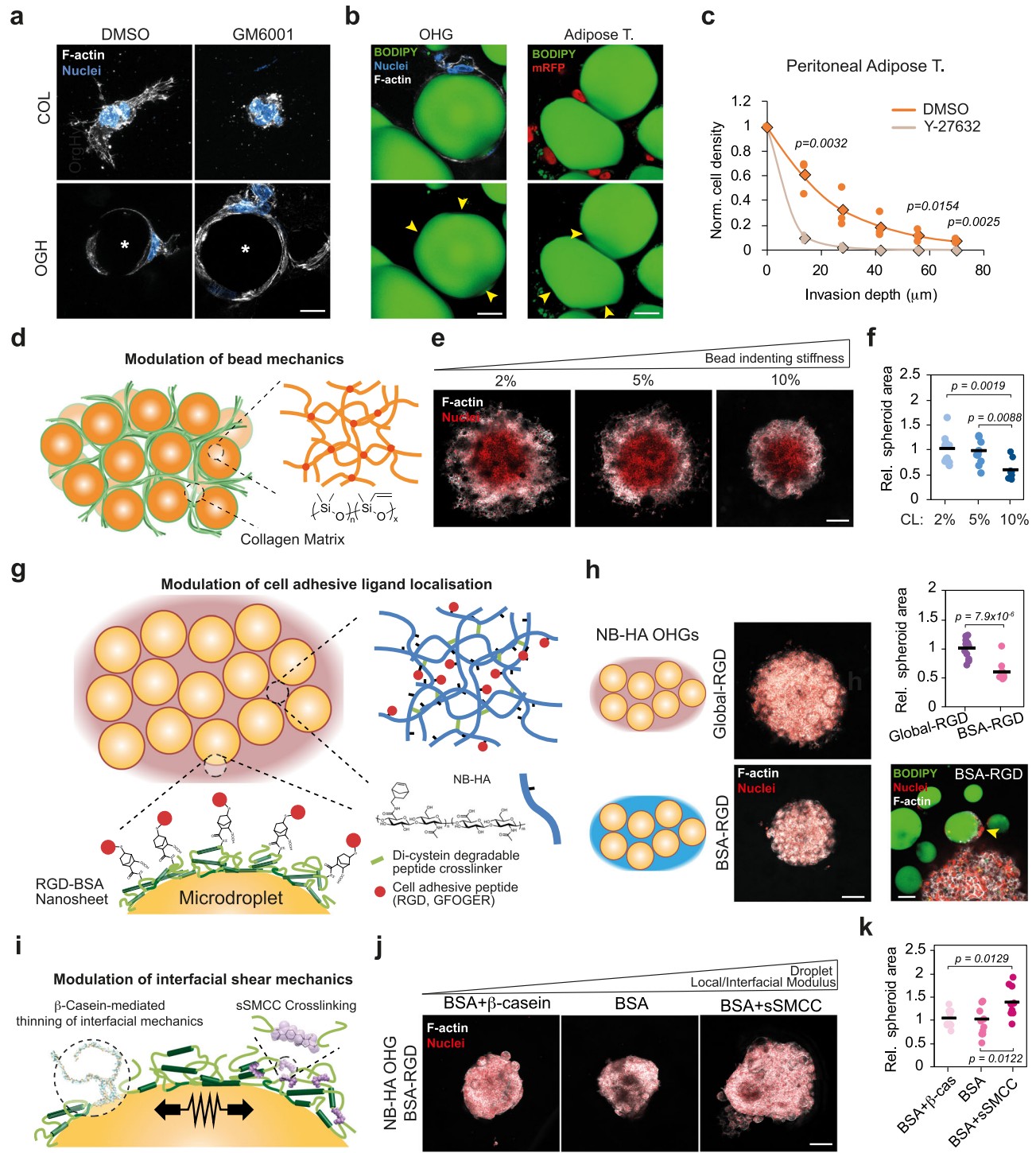

promote single-cell invasion in mesenchymal cells embedded in low-density collagen (where larger pores increase path availability)[30].

Cell confinement is sensed via mechanosensitive ion channels and the deformation of the nucleus, causing reduced nuclear envelope wrinkles[6,31,32]. Indeed, OVCAR8 cells in contact with both 70 and 25 µm microdroplets showed reduced nuclear envelope wrinkles compared to cells embedded in collagen (Supplementary Fig. 14d). Concurrently, proliferation, YAP nuclear localisation, and pMLC expression was similar while pFAK expression was slightly increased in 25 µm compared to 70 µm microdroplet OHGs (Fig. 5j and Supplementary Fig. 14e). These results suggest that cells spreading within OHGs formed with small and large (25 and 70 µm) microdroplets similarly engage the adhesion machinery, but that nuclear deformation is

significantly enhanced in the former OHGs, as a result of the reduced interdroplet distance.

To investigate whether cellular sensors of confinement are activated in OHGs and regulate cell invasion, we first analysed the impact of the mechanosensitive ion channels Piezo1 and TRPV4. Inhibition of Piezo1 did not affect the area of OVCAR8 spheroids embedded in 70 µm OHGs (Supplementary Fig. 15a). In contrast, inhibition of TRPV4 caused a reduction of the spheroid area in both 70 and 25 µm OHGs (Supplementary Fig. 15b). Analysis of spheroids under TRPV4 inhibition revealed that cell invasion into OHGs was not blocked but, as shown previously[33,34], cell proliferation diminished (Supplementary Fig. 15c−e). Next, we investigated whether the cytosolic phospholipase A2 (cPLA2)-Arachidonic acid-InsP3Rs-myosin II signalling cascade,

**Fig. 4 | Microdroplet mechanical anisotropy enables cell invasion. a** F-actin (grey) and DAPI (blue) in OVCAR8 cells embedded in collagen or collagen-based organo-hydrogels (OHG) for 24 h (representative images from $n = 3$ gels). Microdroplet locations are indicated with an asterisk. **b** BODIPY (green), F-actin (grey) and nuclei (blue) staining of OVCAR8 cells embedded in collagen-based OHG (left; $n = 3$ gels) or BODIPY (green), mRFP (red) staining of mRFP-OVCAR8 cells invading into peritoneal adipose tissue (right; $n = 3$ tissues from 1 donor). Arrowheads indicate oil microdroplet (left) and adipocyte (right) deformations at the contact points with cells. **c** Quantification of OVCAR8 organotypic invasion depth into peritoneal adipose tissues after 7 d ($n = 3$ tissues from 1 donor). Rhombuses indicate the average. **d** Schematic representation of the Sylgard 184 PDMS-based OGH. **e** F-actin (grey) and nuclei (red) staining of OVCAR8 spheroids embedded for 7 d in collagen-based OHGs prepared with PDMS microdroplets of varying stiffness (representative images from $n = 10$ spheroids). **f** OVCAR8 spheroid area after 7 d in collagen-based OHGs with PDMS microdroplets prepared with varying crosslinker percentage, relative to average area in 2 wt% microbead OHGs ($n = 10$ spheroids). **g** Schematic representation of the norbornene-functionalised hyaluronic acid (NB-HA) OGH with norbornene-functionalised bovine serum albumin (NB-BSA) used for RGD presentation at the microdroplet surface. **h** F-actin (grey) and nuclei (red) of OVCAR8 spheroids after 7 d in NB-HA OHG with localised presentation of RGD (left). Relative OVCAR8 spheroid area in NB-HA OHG with localised RGD presentation after 7 d in culture ($n = 11$ spheroids; top right). BODIPY (green), nuclei (red), and F-actin (grey) staining of OVCAR8 cells in NB-HA OHG presenting RGD on the microdroplet surface (bottom right). Arrowhead indicates a cell spreading at the microdroplet-HA interface. **i** Schematic representation of the system to modulate microdroplet interfacial mechanics. **j** F-actin (grey) and nuclei (red) of OVCAR8 spheroids after 7 d in NB-HA OHGs (microdroplet-restricted RGD presentation) with varying microdroplet interfacial modulus (representative images from $n = 9$ spheroids). **k** OVCAR8 spheroid area in NB-HA OHGs with microdroplet-presenting RGD and varying protein nanosheet interfacial mechanics after 7 d in culture, relative to BSA-only control ($n = 9$ spheroids). For the data in (**c** and **h**), a two-sided unpaired $t$ test was performed. One-way analysis of variance (ANOVA) with Tukey's correction for multiple comparisons was performed for the data in (**f** and **k**). Scale bars, 25 μm (**a**, **b**), 50 μm (**h**, bottom right), 200 μm (**e**, **h** left, **j**). Source data are provided as a Source Data file.

---

involved in nuclear sensing of confinement, regulates OVCAR8 invasion into OHGs[31,32]. Inhibition of the signalling cascade did not cause major effects on OVCAR8 invasion in either 70 or 25 μm OHGs (Supplementary Fig. 16a). Conversely, activation of the cascade with arachidonic acid reduced spheroid area and invasion (Supplementary Fig. 16b, c). Therefore, inhibition of previously described mechanosensors activated in confinement were not found to regulate OVCAR8 cell invasion into OHGs.

Finally, we investigated whether a mesenchymal phenotype, inducing single cell migration, facilitates invasion into 25 μm OHGs at 80% oil volume fraction. Indeed, we observed that CAOV3 cells migrate collectively and present reduced invasion in 25 compared to 70 μm OHGs (Supplementary Fig. 17a, b). To induce a mesenchymal phenotype, we treated CAOV3 cells with TGFβ. TGFβ treatment caused enhanced invasion in both 70 and 25 μm OHGs (Fig. 2j, k and Supplementary Fig. 17c). However, the impact of TGFβ treatment on enhancing CAOV3 invasion was greater in 25 than in 70 μm OHGs (Fig. 5k, l). Therefore, these data indicate that the biochemical microenvironment, through TGFβ-induced phenotypical changes in cancer cells, synergises with the biophysical alterations occurring in adipose tissues (i.e., reduced adipocyte size) to regulate ovarian cancer cell invasion.

## Discussion

Biomimetic OHGs recapitulate the architecture and mechanical characteristics of native adipose tissues at the nano- to macro-scale, and enable the direct encapsulation of cells and spheroids, or the study of cell migration in invasion assays. OHGs recreated the invasive behaviour of ovarian cancer cells observed in peritoneal adipose tissues, which collagen gels could not replicate. Similarly, although previous hydrogel models have mimicked the mechanical and biochemical properties of the tissue ECM, the greater complexity of native tissues, including adipose tissues, can lead to discrepancies between in vitro and in vivo cell migration studies[35]. Granular hydrogels and other porous hydrogels introduce structural features of tissues such as interstitial spaces and porosity, demonstrating that such characteristics can regulate cell invasion[36–39]. Similarly, the size and shape of the particles forming granular hydrogels can control cell invasion[39]. Here, we demonstrate that by including both ECM-like and adipocyte-like phases in OHGs, ovarian cancer cell invasion can be induced without the requirement of pro-invasive adipose tissue-derived biochemical factors.

We find that cells do not require MMP activity to invade adipose tissues and OHGs. Instead, the establishment of adhesion and myosin-regulated migration tracks at the adipocyte-ECM interface enables cell invasion (Fig. 5m). However, MMP expression is associated with poor overall survival in HGSC, possibly due to their function in breaching the mesothelial layer and basement membrane, and in regulating the subsequent fibrotic remodelling of the tissue[40–44]. We show that ECM adhesion at the adipocyte-ECM interface is fundamental for cell invasion. Integrin-mediated adhesion to collagen, FAK activity, and YAP are also involved in therapy resistance in ovarian cancer[43]. Thus, various clinical trials targeting the ECM and related signalling in ovarian cancer are currently underway[45].

Cells migrating at the adipocyte/microdroplet-ECM interface are under confinement, especially in tissues and OHGs containing small diameter adipocyte/microdroplet and low ECM volume fraction, restricting cell invasion. This implies that larger adipocytes, as observed in women with high body mass index, a factor associated with poorer survival, are more permissive to cell invasion[46,47]. Although we observed cellular hallmarks of confinement, such as nuclear deformations and nuclear envelop stretching, cells did not present enhanced DNA damage, and the inhibition of confinement-related pathways had a minimal effect on invasiveness[31,32]. This suggests that confinement does not reach the threshold to induce these effects, possibly facilitated by the mechanical anisotropy of adipocytes/microdroplets (high compliance radially, but displaying high interfacial shear moduli), leading to the opening of migration tracks. In addition, the induction of phenotypic plasticity and a mesenchymal phenotype[48,49], as well as cancer-associated stromal cells[50–52], may facilitate invasion into OHGs, particularly within smaller microdroplets, and into adipose tissues. Whether these mechanisms are unique to ovarian cancer cells remains unknown, but understanding them will be crucial to elucidating how other cell types, including immune cells, stromal cells, and other cancer cells, navigate adipose tissues.

In this study, we did not explore stromal-cancer cell communication, however, we envision OHGs will be fundamental to fully understand this process. Indeed, using patient-derived cells, we observed that PAX8- stromal cells present higher invasion and proliferation in OHGs than in microdroplet-free collagen gels, a phenomenon that may be fundamental to understand the evolution of omental and peritoneal metastases. Since adipose tissues are involved in multiple cancer types, including gastric, breast, colon, pancreatic, renal cancers and leukaemia[53], adipose tissue models such as OHGs will be central for decoding the complex regulation of ECM and cell communication in these tissues.

## Methods

### Ethical statement

All participants in this study provided written informed consent, and the study was approved by the Swedish Ethical Review Authority (Dnr: 2019-05149 and 2023-04696-01). All procedures were performed in

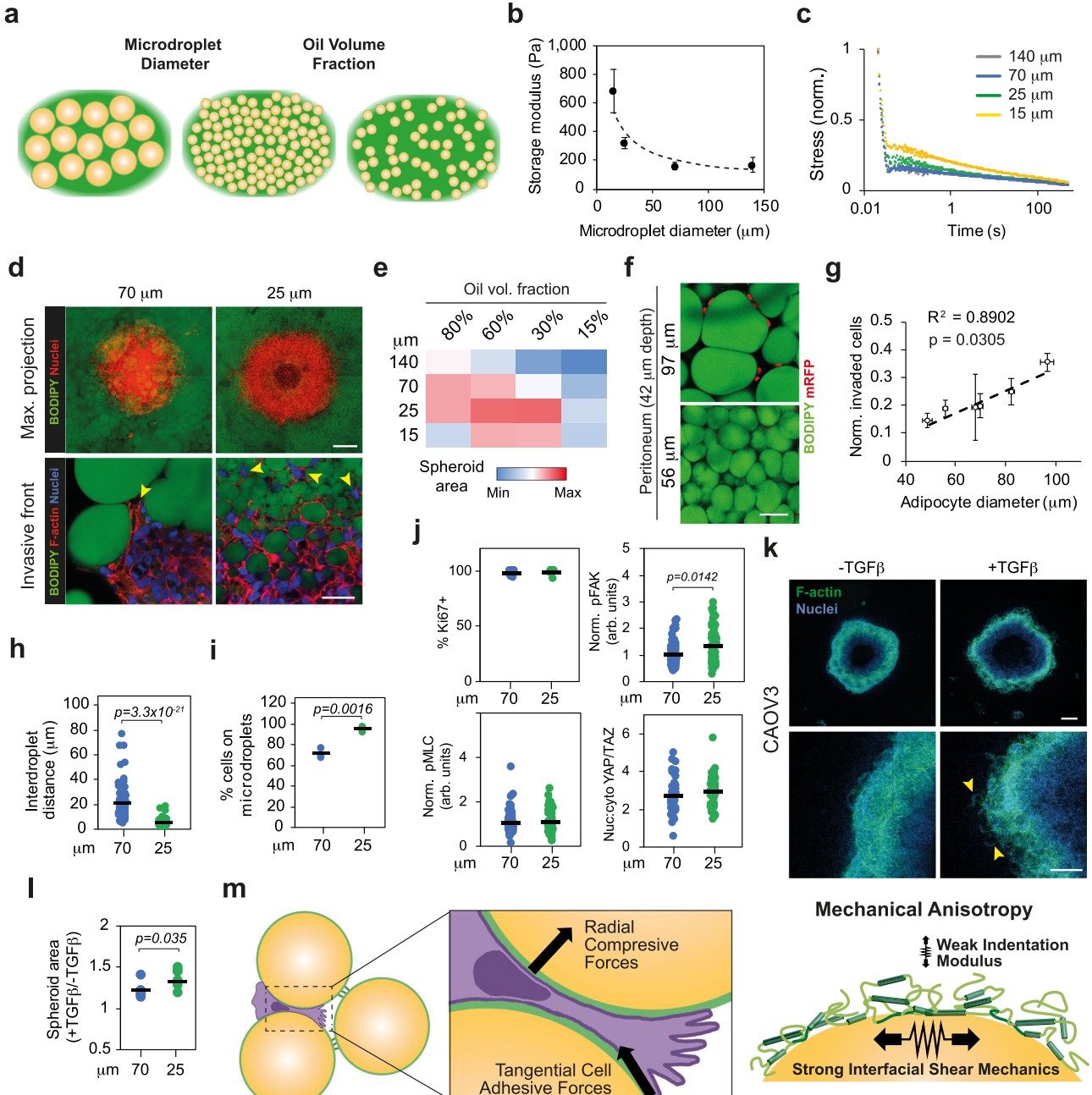

**Fig. 5 | Microdroplet size and volume fraction regulate invasion and migration mode. a** Schematic representation of the organo-hydrogels (OHGs) with varying microdroplet diameter and volume fraction. **b** Storage modulus of collagen-based OHG with distinct microdroplet size ($n = 4$ gels; average ± s.e.m.). The dashed line shows the predicted trend. **c** Normalised stress-relaxation curves of collagen-based OHG with distinct microdroplet size ($n = 4$ gels). **d** BODIPY (green) and F-actin (red) and nuclei (blue) staining of OVCAR8 cells after 7 d in collagen-based OHG of 70 or 25 μm diameter microdroplets (representative images from $n = 13$ spheroids). Arrowheads indicate cells at the invasive front. **e** Quantification of relative OVCAR8 spheroid area in collagen-based OHG of distinct emulsion percentage of volume fraction and microdroplet diameter after 7 d in culture ($n = 13$ spheroids). **f** BODIPY (green) and mRFP (red) staining of mRFP-OVCAR8 cells after 7 d invasion into adipose peritoneal tissue explants (representative images from $n = 3$ explants per donor). **g** Correlation between patient average adipocyte diameter and invaded OVCAR8 cells after 7 d, at 42 μm tissue depth, relative to number of cells at the tissue surface ($n = 6$ donors; average ± s.e.m.). The dashed line shows the predicted trend. **h** Quantification of interdroplet distance in OHG ($n = 100$ microdroplets). **i** Quantification of the percentage of invasive front

OVCAR8 cells in contact with microdroplets in OHG ($n = 121$ cells in 3 gels). **j** Quantification of percentage of Ki67 + cells, immunofluorescence signal intensity of cytoplasmic pFAK and pMLC relative to collagen average, and nuclear:cytoplasmic ratio of YAP/TAZ in OVCAR8 cells embedded in OHG of 70 or 25 μm diameter microdroplets for 24 h ($n = 55$ cells). **k** F-actin (green) and DAPI (blue) staining of CAOV3 spheroids embedded for 7 d in collagen-based 25 μm microdroplet OHG in the presence or absence of TGFβ. Arrowheads indicate invading cells (representative images from $n = 6$ spheroids). **l** Quantification of CAOV3 spheroid area relative to non-TGFβ-treated control after 7 d in culture ($n = 6$ spheroids). **m** Schematic illustration of the cell force-dependent invasion of adipose tissue enabled by the anisotropic mechanics of OHGs and the generation of migration tracks at the ECM-adipocyte/microdroplet interface. Arrows indicate the direction of the force. For the data in (**h, i, j** and **l**), a two-sided unpaired $t$ test was performed. Coefficient of determination and Pearson correlation (two-tailed test) were performed in (**g**) to determine the relationship between tissue adipocyte diameter and OVCAR8 invasion. Error bars in (**b**) represent the s.e.m. Scale bars, 50 μm (bottom panels in **d, k**), 100 μm (**f**), 200 μm (top panels in **d, k**). Source data are provided as a Source Data file.

accordance with relevant guidelines and regulations and adhered to the principles of the Declaration of Helsinki. Study participants received no compensation.

## Patient samples

Patient samples for this study were collected in the operating theatre immediately after incision during cytoreductive surgery in patients with advanced ovarian cancer at Karolinska University Hospital, Stockholm, Sweden. The study samples included ascitic fluid, as well as omental and peritoneal tissues. Whenever possible, the surgeon resected a portion of the peritoneum that appeared free of visible carcinomatosis for collection. The peritoneal samples were resected with and without visible underlying adipose tissue. A reference gynaecologic pathologist at Karolinska University Hospital conducted a centralised review of the final pathology for all included patients. Sex was determined based on medical records and clinical diagnosis of ovarian cancer, which inherently implies assignment as female at birth. Gender identity was not collected, and no gender-based analyses were conducted.

## Human cell line culture

The human cell lines were cultured at 37 °C in 5% $CO_2$. OVCAR3 (HTB-161, ATCC), OVCAR8 (305383, Cytion), OVCAR4 (SCC258, Sigma), Kuramochi (JCRB0098, JCRB) and CAOV3 (HTB-75, ATCC) cells were grown in RPMI medium, while Tyk-nu (JCRB0234.0, JCRB) and Tyk-nu.CPR (JCRB0234.1, JCRB) were cultured in Minimum Essential Medium (MEM). All culture media were supplemented with 10% FBS (v/v) and 100 mg mL$^{-1}$ penicillin/streptomycin. OVCAR8-mRFP and Tyk-nu-mRFP were generated by transfecting OVCAR8 and Tyk-nu cells with recombinant histone H2A-red fluorescent protein and subsequently sorting highly expressing cells by flow cytometry. Nuclear mRFP expression was confirmed by confocal microscopy.

## Primary human cell culture

Patient ascites fluid was surgically drained from the peritoneal cavity and collected at the operating theatre of Karolinska University Hospital prior to ultra-radical hysterectomy of HGSOC tumours. The transportation and processing of the fluid was conducted in sterile conditions. Isolation of cells from ascites fluid was performed immediately after acquisition. After centrifugation at $800 \times g$ for 10 min, the supernatant was filtered using a 0.20 μm bottle filter (Thermo Scientific™ Nalgene™ Rapid-Flow™) to obtain cleared fluid used as a supplement for cell culture media. If needed, red blood cells were lysed from the cell pellet by using Tris-buffered ammonium chloride solution (Tris-NH4Cl). Multicellular clusters were collected from the cell suspension using 45 μm strainer and re-suspended in 1:1 DMEM:F12 medium with 100 mg mL$^{-1}$ penicillin/streptomycin and 10% clarified ascites. Cells were cultured at 37 °C in a 5% $CO_2$ incubator.

## Human explant culture

Fresh macroscopically normal peritoneal tissues were cut into 1-2 mm$^3$ pieces and placed into non-adherent U-bottom 96 well plates (ThermoFisher Scientific). A solution with 10,000 cells in 10 μl media was added, and cells were allowed to attach to the tissues for 1 h at 37 °C. Then, 150 μl media was added. Tissues were cultured for 7 days prior to overnight fixation with 4% paraformaldehyde (PFA). For quantification, tissues were imaged from the surface to the core (z-stacks). Cell invasion depth was analysed by quantifying the number of mRFP + nuclei in each stack. Data is presented as the number of cells in each stack, representing a specific depth from the surface, relative to the number of cells at the surface.

## Cell invasion assays

For the spheroid invasion assay, 3000 cells were seeded into U-bottom ultra-low attachment 96-well plates (ThermoFisher Scientific) and centrifuged at $200 \times g$ for 5 min. Spheroids/Clusters were allowed to form over 4 days in culture. Single spheroids were transferred into non-gelled collagen or OHG solutions, and hydrogels were formed. Cells were incubated for 0 or 7 days prior to fixation with 4% PFA. For quantification, spheroids were imaged, and the area of the outline of their maximum-intensity projection image was quantified with ImageJ (v 2.14.0/1.54 f). The spheroid area presented is normalised to control conditions.

For the organotypic invasion assay, 15 μl OHG or collagen were pre-formed in U-bottom ultra-low attachment 96-well plates. Subsequently, 150 μl media containing 10,000 cells was added to each well. Cells were allowed to invade the hydrogels for 7 days prior to fixation with 4% PFA. For quantification, hydrogels were imaged from the surface to the core (z-stacks). Cell invasion depth was analysed by quantifying the number of mRFP + nuclei in each stack. Data is presented as the number of cells in each stack, representing a specific depth from the surface, relative to the number of cells at the surface.

## BSA functionalization

Norbornene-functionalised BSA (NB-BSA) was prepared using the following protocol. 10 g of BSA (15 mmol, Sigma-Aldrich) were dissolved in 100 mL of carbonate (NaHCO$_3$/Na$_2$CO$_3$) buffer at pH 9. Then, 2 g of carbic anhydride (12.2 mmol, ThermoFisher Scientific) were slowly introduced into the reaction mixture at room temperature. The resulting mixture was then heated whilst stirring at 40 °C for 4 h. A pale-yellow solution was recovered, dialysed against deionised water for 4 days and freeze-dried. The functionalisation degree of the resulting NB-BSA was determined by $^1$H NMR, by integration of the 2 alkene protons (δ 6.0- 6.3 ppm) of norbornene moieties, normalised by the aromatic protons of BSA (δ 6.5-7.5 ppm).

## Oil-in-water emulsions

To generate adipocyte-mimicking microdroplets, emulsions of silicone oil droplets (10 cP, Sigma) stabilised with BSA or NB-BSA were prepared. First, 2 mL silicone oil was added into 10 mL PBS solutions containing 10% BSA (w/v). To obtain microdroplets stabilised by protein assemblies with softer interfacial shear moduli, emulsions were prepared with 9.9% BSA, combined with 0.1% (w/v) β-casein (Sigma). Solutions were then vortexed or homogenised to generate the corresponding emulsions, varying the homogenisation speed and time to control the size of the resulting oil microdroplets. The emulsions were then centrifuged at $300 \times g$ for 5 min. After the remaining BSA solution was removed, the emulsions were washed three times with 10 mL PBS. The emulsions were centrifuged twice at $200 \times g$ for 2 min to remove excess PBS. To generate microdroplets stabilised by protein assemblies with higher interfacial shear storage moduli, BSA-formed emulsions were crosslinked with 2 mg mL$^{-1}$ sulfo-succinimidyl 4-(N-maleimidomethyl) cyclohexane-1-carboxylate (sulfo-SMCC, Thermofischer Scientific) for 1 h at RT and washed 3x with PBS.

## Preparation of PDMS Microdroplets

Polydimethylsiloxane (PDMS) emulsions containing bovine serum albumin (BSA) were prepared using Sylgard 184 silicone base and crosslinker in various proportions. To generate PDMS emulsions, 2 g of Sylgard 184 base were mixed with 0.04 g (2%), 0.1 g (5%), or 0.2 g (10%) PDMS crosslinker. Each mixture was gently stirred to ensure homogeneity. The tubes were then placed under vacuum, with the lids open, for 30 min to remove trapped air bubbles. Following degassing, bovine serum albumin (BSA) was dissolved in phosphate-buffered saline (PBS) to a concentration of 100 mg mL$^{-1}$. A total of 8 mL of this BSA solution was added to each PDMS mixture. To form the emulsions, the samples were homogenised using a T25 Ultra Turrax Homogeniser at a speed of 12,000 rpm for 5 min, ensuring that the homogeniser probe fully immersed in the oil phase during the process. After homogenisation,

the samples were centrifuged at 1000 x $g$ for 5 min to remove any air bubbles that formed during mixing.

The samples were then heated in an oil bath at 50 °C for 4 h to cure the PDMS. After heating, each sample was vortexed to resuspend the emulsion microdroplets and subsequently centrifuged at 300 x $g$ for 5 min. Excess aqueous solution was carefully removed using a pipette. To sterilise the samples, 5 mL of a 5 mg mL$^{-1}$ sodium azide solution in PBS was added to each tube, and the samples were stored at 4 °C overnight. The following day, the samples were vortexed and centrifuged again at 300 x $g$ for 5 min. The supernatant was discarded, and the emulsions were washed 3X by adding 5 mL of sterilised PBS. After the final wash, any remaining liquid was removed using a pipette, leaving behind the final PDMS microbeads.

### Collagen-based hydrogels
To prepare collagen-only and collagen-based organohydrogels, a 2.25 mg mL$^{-1}$ solution of type I collagen (C7661, Sigma) was first prepared. This solution contained a 1:1 ratio of the collagen stock (4.5 mg mL$^{-1}$ in 0.4% (v/v) Acetic acid) and 2x minimum essential medium (MEM). The pH of the solution was then adjusted to pH 7 with cold 1 M NaOH. For the organohydrogels, 20% volume of the prepared collagen was mixed with 80% volume of the undiluted emulsion, giving a final aqueous phase concentration of collagen of 1.125 mg mL$^{-1}$. To obtain an equal concentration of collagen (1.125 mg mL$^{-1}$) for the collagen-only gels, the collagen solution was mixed 1:1 with cold PBS. For cell embedding, the prepared solutions were mixed with cells. Then, the solutions were added into plates and placed in a humidified incubator at 37 °C for 1 h. After incubation, media were added. To analyse collagen concentration and microarchitecture, cell-free collagen gels and collagen-based OHGs were fixed with 4% PFA, stained for collagen-I, and imaged at 63x magnification. Collagen density and pore size were quantified by ImageJ (v 2.14.0/1.54 f).

### HA-based organo-hydrogels
To functionalise hyaluronic acid (HA) with norbornene, sodium hyaluronate (0.2 g, 0.527 mmol for each repeating unit) (Mw: 200,000 g mol$^{-1}$) was dissolved in 20 mL of MES buffer (0.1 M, pH 5.5). The solution was purged with nitrogen, and then EDC-HCl (0.148 g, 0.772 mmol) and NHS (0.089 g, 0.772 mmol) were added in one portion and left to stir for 10 min. Subsequently, the pH of the solution was increased to approximately 8 using NaOH (2 M). 5-norbornene-2-methylamine (0.1 mL, 0.812 mmol) was added by syringe to the reaction solution while stirring under a nitrogen atmosphere and was left to stir overnight at room temperature. NaCl (0.75 g, 12.83 mmol) was then added to the reaction solution to neutralise the reaction product and form sodium hyaluronate derivatives. The reaction solution was then precipitated into a 10-fold excess cold (4 °C) acetone. The precipitate was dissolved in 20 mL of deionised water and dialysed against deionised water by changing the dialysate every 24 h for 5 days. Finally, the solution was frozen in liquid nitrogen and lyophilised to yield norbornene-functionalised hyaluronic acid (NB-HA). The degree of functionalisation of NB-HA derivatives was determined via 1H NMR spectroscopy. 1H NMR shifts of attached norbornenes (D2O) are δ: 6.33 and 6.02 ppm (vinyl protons, endo), 6.26 and 6.23 ppm (vinyl protons, exo). To calculate the degree of substitution (DS), the integrations of these peaks were normalised to the peak of the methyl group of HA monomer at δ: 2.0 ppm.

To form HA-based organohydrogels, 80% (v/v) emulsions were mixed with 20% (v/v) 3.75 % (w/v, in PBS) NB-HA and cells or spheroids, and crosslinked with 1.3 mM PEG di-thiol (Mw 1000 g mol$^{-1}$; Sigma) or 1.3 mM GCGPQGIAGQGCG (MMP-cleavable, Proteogenix, > 95% purity, TFA removed), 1 mM lithium phenyl-2,4,6-trimethylbenzoylphosphinate (LAP; Sigma) and ultraviolet (UV) light for 30 s. For cell adhesion, 2 mM of CGGRGDSP*K*-FITC-G (RGD-FITC, Proteogenix, > 95% purity, TFA removed), GCGGRGDSPG (RGD,

Proteogenix, > 95% purity, TFA removed), or trimerised (Ac)-GGYGGGPGGPPGPPGPPGPPGPPGPPGFOGERGPPGPPGPPGPPGPPGPC-(Am) (GFOGER, Proteogenix, > 95% purity, TFA removed) were added to the solution or to NB-BSA emulsion, prior to UV light exposure. For localised peptide presentation (only at the surface of microdroplets), emulsions were washed 3X with PBS following photo-irradiation in the presence of 2 mM RGD or RGD-FITC in PBS and were then mixed with the crosslinker, LAP and cells, in the absence of additional adhesion ligands.

### Antibodies
Primary antibodies: Ki67 (1:400, 9449S, Lot. 12, Cell Signalling Technology) and cleaved Caspase-3 (1:800, 9664, Lot. 22, Cell Signalling Technology), Collagen-I (1:800, MA1-26771, Lot. XA3474955, Thermo Fisher Scientific), pFAK (1:200, phosphorylated (Y397) FAK, 611807, Lot. 6092601, BD Biosciences), pMLC (1:400, phosphorylated (S20) myosin light chain, ab2480, Lot. GR3241064-7, Abcam), YAP/TAZ (1:200, sc-101199, Lot. J3013, Santa Cruz Biotechnology), γH2A.X (1:200, phosphorylated (S139) gamma H2AX, #2577S, Lot.11, Cell Signalling Technologies), LAP2 (1:400, PA5-52519, Thermo Fisher Scientific), PAX8 (1:400, 10336-1-AP, Lot. 00099975, Thermo Fisher Scientific), CK7 (1:400, MA5-11986, Thermo Fisher Scientific). Secondary antibodies: Alexa Fluor Plus 488 anti-mouse IgG (1:1000, A-21202, Lot. VC300588, Thermo Fisher), Alexa Fluor 350 anti-rabbit IgG (1:500, A10039, Lot. 2533800, Thermo Fisher) and Alexa Fluor Plus 555 anti-rabbit IgG (1:1000, A32794, Lot. VI311611, Thermo Fisher), Alexa Fluor 647 anti-rabbit IgG (1:500, A31573, Lot. 1826679, Thermo Fisher Scientific), Alexa Fluor 647 anti-mouse IgG (1:500, A31571, Lot. 1900251, Thermo Fisher Scientific).

### IF of hydrogels
Cells were fixed with 4% paraformaldehyde (PFA) for 1 h. For nuclear proteins (YAP/TAZ, Ki67, γH2A.X, and LAP2), cells were post-fixed for 1 min with a 1:1 solution of ice-cold acetone-methanol. Before incubation with primary antibodies, the cells were blocked in blocking buffer (15% FBS and 0.3% Triton-X in PBS) for 2 h at RT. Next, samples were incubated overnight at 4 °C with primary antibodies diluted in blocking buffer. Subsequently, the samples were washed twice with PBS containing 0.45% Triton-X and stained with the corresponding fluorescence conjugated secondary antibodies for 2 h. In addition, some samples were also incubated with 10 µM BODIPY 493/503 (in DMSO; Thermo Fisher Scientific) and Phalloidin Alexa Fluor 647 or 488 (1:400, Thermo Fisher Scientific) in blocking buffer for 2 h at RT. Lastly, the cells were washed once with 0.45% Triton-X and once with PBS. Gels were clarified with 40% Glycerol (Sigma) in PBS to match the refractive index of the oil.

### IF of tissue explants
Cells were fixed with 4% PFA overnight at 4 °C. Samples were quenched with 0.3 M Glycin in PBS for 5 min at RT and washed twice with PBS. Samples were blocked for 1 h at RT with 2% FBS in 0.4% Triton X-100 in PBS. Samples were incubated with the primary antibody diluted in blocking buffer overnight at 4 °C. Then, samples were washed for 5 min once with 0.1% PBS-Tween and twice with PBS. Next, samples were incubated in secondary antibody, 10 µM BODIPY (Thermo Fisher), Hoechst 33342 (1:1000), and Phalloidin Alexa Fluor 647 (1:400) overnight at 4 °C. Subsequently, samples were washed for 5 min once with 0.1% PBS-Tween and twice with PBS and mounted in Omnipaque for tissue clearing.

### Confocal fluorescence microscopy
Hydrogel and tissue samples were transferred into CultureWell gaskets adhered to a glass coverslip. Images were acquired with 10x Air, 20x Air, and 63x Oil objectives in a Zeiss LSM 800 Airy or a Zeiss LSM 900 Airy laser-scanning confocal microscope operated by the ZEN Blue

v2.6 software. Image analysis was performed on Fiji-ImageJ (v 2.14.0/1.54 f).

## Microdroplet and adipocyte shape analysis

For the analysis of microdroplet/bead/adipocyte deformation, the shape of BODIPY-stained microdroplets/beads/adipocytes in close proximity with mRFP + nuclei or those without added ovarian cancer cells that were not in direct contact with other microdroplets/beads/adipocytes (as these can cause deformation of neighbouring microdroplets/beads/adipocytes), were quantified by ImageJ (v 2.14.0/1.54 f).

## Rheology

Rheological measurements were carried out using a hybrid rheometer (DHR-3; TA Instruments). Collagen-based OHGs (0.5 mL) were pre-formed and incubated for 1 h at 37 °C prior to transfer to the rheometer. Samples were placed between an upper parallel plate geometry (20 mm) and a standard Peltier plate pre-heated at 20 °C. Then, the geometry was lowered to the desired gap. First, frequency sweeps were carried out, recording the storage modulus (G') and loss modulus (G'') as a function of frequency in a range of 0.1–100 Hz, at 1% strain. Amplitude sweeps from 0.1 to 100 % strain were subsequently carried out, at an oscillating frequency of 1 rad/s. Stress relaxation was performed at 2% strain for 120 s. G' reported corresponds to those measured at 10 rad s$^{-1}$.

Photorheological measurements were performed using a hybrid rheometer (DHR-3) fitted with a UV accessory. 100 µL solutions of hydrogel precursors were placed between an upper parallel standard Peltier plate geometry (20 mm) and a bottom quartz window (allowing transmittance of UV light) at a fixed gap of 250 µm. UV irradiation (Omnicure S1500 mercury lamp, λ 280 – 600 nm, connected via a light guide and calibrated with an ILT 1400-A radiometer) started after a period of oscillation (30 s) to equilibrate the system. To identify the gelation point and monitor the crosslinking progression, time sweep measurements (10 min) at a controlled strain of 0.4% and a constant angular frequency of 1 Hz (6.28 rad/s) were performed at room temperature. Storage modulus (G') and loss modulus (G'') were recorded as a function of time in TRIOS software. Subsequently, a frequency sweep was carried out, recording G' and G'' as a function of frequency in a range of 0.1–100 Hz at 0.4% strain. Amplitude sweeps were performed at an oscillating frequency of 1 Hz. Finally, stress relaxation measurements were carried out and gels were subjected to 2.0% strain, which was reached in 2 s and held for 180 s.

## Interfacial rheology

Interfacial shear rheological measurements were carried out on a Discovery Hybrid-Rheometer (DHR-3) from TA Instruments, using a Du Nouy ring geometry and a Delrin trough with a circular channel. The Du Nouy ring has a diamond-shaped cross-section that improves positioning at the interface between two liquids to measure interfacial rheological properties while minimising viscous drag from upper and sub-phases. The ring has a radius of 10 mm and is made of a platinum −iridium wire of 400 µm thickness. The Derlin trough was filled with 4 mL of fluorinated oil (Novec 7500, ACOTA). Using axial force monitoring, the ring was positioned at the surface of the fluorinated oil and was then lowered by a further 200 µm to position the medial plane of the ring at the fluorinated phase interface. 4 mL of the PBS solution was then gently introduced to fully cover the fluorinated sub-phase. Time sweeps were performed at a frequency of 0.1 Hz and a temperature of 25 °C, with a displacement of 10$^{-3}$ rad (strain of 1%) to follow the self-assembly of the proteins at corresponding interfaces. 15 min after the beginning of the time sweep, the protein solution (100 mg mL$^{-1}$ in PBS) was added in the aqueous phase to a final concentration of 1 mg mL$^{-1}$. In the case of non-crosslinked albumin interfaces, the time sweep was carried out for 3 additional hours.

In the case of crosslinked albumin nanosheets, the remaining soluble proteins were removed after 1 h by washing the aqueous phase for 30 min with a microfluidic flow controller (OB1 MK4 pressure controller ElveFlow, Elvesys), which allowed the regulation of inlet and outlet PBS flow rates (overall 1 mL min$^{-1}$). Following this washing cycle, a sulfo-SMCC solution was injected in the aqueous phase to a final concentration of 2 mg mL$^{-1}$. The excess crosslinker was removed by a 30 min washing cycle after 1 h.

In both crosslinked and non-crosslinked systems, frequency sweeps (with displacements of 10$^{-3}$ rad) and amplitude sweeps (at a frequency of 0.1 Hz) were carried out at 25 °C to examine the frequency-dependent behaviour of corresponding interfaces and to ensure that the selected displacements and frequencies were within the linear viscoelastic region.

## Atomic force microscopy

To perform AFM nanoindentation, samples were placed in a 35 mm petri dish and submerged in PBS. Force–distance measurements were carried out on a JPK NanoWizard 4 (JPK Instruments), with samples submerged in PBS at room temperature. Pyrex-nitride colloidal probes (diameter 15 µm; CP-PNPL-SiO-E-5, NanoAndMore) were used (spring constant (K) was calibrated, 0.08 N m$^{-1}$). Scans of 10 × 10 µm grids of indentation at three different locations across the surface of two different samples were acquired for each condition, and force-displacement curves were recorded. Indentations were carried out with a relative setpoint force of 2 nN and a loading rate of 5 µm s$^{-1}$. Data were collected using JPK proprietary SPM software (JPK Instruments), and Young's moduli were calculated using the Hertz model for a spherical tip (Matlab, v9.14).

## Small-molecule inhibitors, agonists, and growth factors

The pharmacological and small-molecule inhibitors and agonists used were the following: rat anti-human CD29 (1:100; BD Bioscience) to block integrin β1, blebbistatin (1 µM; Enzo Life Sciences) for myosin II inhibition, GM6001 (10 µM; Calbiochem) as broad-spectrum MMP inhibitor, Y-27632 (10 µM; MedChemExpress) as ROCK inhibitor, Defactinib (10 µM; MedChemExpress) as FAK inhibitor, verteporfin (1 µM; Sigma) as YAP/TAZ inhibitor, arachidonic acid (70 µM; MedChemExpress) to mimic signalling downstream of nuclear mechanosensing, AACOCF3 (28 µM; Tocris) to inhibit signalling downstream of nuclear mechanosensing, 2-aminoethoxydiphenylborane (2APB; 20 µM; MedChemExpress) as IP3 receptor agonist, Xestospongin c (10 µm; Tocris) as IP3 receptor agonist, GSMTx4 (10 µm; MedChemExpress) as Piezo 1 inhibitor, GSK205 (10 µm; MedChemExpress) as TRPV4 inhibitor, and GSK1016790A (50 µm; MedChemExpress) as TRPV4 agonist. The human recombinant TGFβ (1 ng mL$^{-1}$; R&D Systems) was used to induce epithelial-mesenchymal transition.

## Bioinformatics

Gene expression data of human cell lines were obtained from publicly available datasets including the Gene Expression Omnibus (GEO) GSE50831 and GSE28724 datasets. The RNA raw data were normalised and compared using the Affymetrix Expression Console Software (v 4.0.1; Thermofisher). Gene Set Enrichment Analysis (GSEA, v 4.3.2) was performed with significantly upregulated and downregulated genes ( > 2-fold; p < 0.05).

## Statistical analysis

Measurements were performed on at least 3 biological replicates from separate experiments. Statistical tests performed and sample size (n) for each figure are indicated in the figure legends. Values of $p < 0.05$ were considered statistically significant. Data were handled and analysed in Microsoft Excel (v 16.16.27) and GraphPad Prism (v 10.4.1). The bar plots indicate the mean, and the error bar represents the standard deviation, unless otherwise indicated in the figure legends. The

individual data points are also represented as circles. In other data points, the mean is represented as a line and individual data points are indicated as circles. In the box plots, boxes represent the interquartile range, the centre of the boxes represents the median, and the cross represents the mean. The whiskers represent the maximum and minimum values, excluding the outliers, which are represented as circles.

## Reporting summary

Further information on research design is available in the Nature Portfolio Reporting Summary linked to this article.

## Data availability

The data generated in this study are provided in the Supplementary Figures. And the Source Data file. The gene expression data of human cell lines used in this study are available in the Gene Expression Omnibus (GEO) repository under the accession codes GSE50831 and GSE28724. Source data are provided in this paper.

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

## Acknowledgements

We thank the Biomedicum Imaging Core at Karolinska Institutet for imaging facilities. We thank early work and insight of Dr William Megone on the mechanical anisotropy of liquid-liquid interfaces. The authors thank the European Research Council (ProLiCell, 772462; ProBioFac, 966740 to J.E.G.), the Norwegian Cancer Society (216113 to K.L.), the Novo Nordisk Foundation (NNF21OC0070381 to K.L.), the Swedish Cancer Society (21 1888 Pj to K.L.; 20 0245 P 03H to S.S.; 230627 JCIA to S.S), the Region Stockholm County Council (20200004 to S.S.), the Cancer Research Funds of Radiumhemmet (194142; 224093 to S.S.) and the Swedish Society of Medicine (972109; 998436 to S.S.) for support. Y.Z. thanks the China Scholarship Council for a studentship (202208060332). William Megone and the Engineering and Physical Sciences Research Council (EP/L505602/1).

## Author contributions

J.G.-M., F.B. and J.E.G. designed the research. J.G.-M., D.M., Y.Z., C.N. and A.C. synthesised and characterised materials and tissues. J.G.-M., P.N., S.A. and M.F.R. carried out experiments. J.G.-M., P.N., D.M., Y.Z., C.N. and A.C. carried out data analysis. O.G. and S.S. extracted and processed human patient samples. J.G.-M., T.A., K.L. and J.E.G. supervised the work. J.G.-M. and J.E.G. co-wrote the manuscript. All authors edited the manuscript and approved its final version.

## Competing interests

The authors declare no competing interests.
