## [Transparent Peer Review file · Nature Communications]

Biomimetic organo-hydrogels reveal the adipose tissue local mechanical anisotropy regulates ovarian cancer invasion

Corresponding Author: Dr Jordi Gonzalez-Molina

Version 0:

Reviewer comments:

Reviewer #1

(Remarks to the Author)

This manuscript by Gonzalez-Molina and coworkers describes the generation and characterization of biomimetic organo-hydrogels to mimic the mechanical properties of visceral adipose tissues and characterize the behavior of ovarian cancer (OC) cells in these gels in comparison to adipose tissue in attempt to delineate the tropism of metastasizing OC cells to adipose. Mechanistic studies implicate a role for TGF β in this process. While the study is of interest, several points should be addressed.

Major points:

1. The rationale for the design of the organo-hydrogels (OHGs) requires further information. It is shown in Fig. 1 that 70 μ m silicone oil microdroplets were used to mimic visceral adipocytes. Additional information is required on the system that is being mimicked. The methods just indicate 'macroscopically normal omental and peritoneal tissues' were examined. Presumably these tissues are from OC patients and are therefore unlikely to be normal and to not already contain OC cells (particularly the omental tissue). If peritoneal tissue, where, specifically, in the peritoneum were the tissues obtained? Were tissues stained for OC cells? Some of the data in Extended Fig. 1 would be better incorporated into Fig. 1, particularly data showing similar diameter. However this figure needs additional detail as well. It is indicated that n=170 cells were measured to determine diameter. From how many patients were these values obtained? Was patient age or overall adiposity (body mass index or similar) taken into account? A more clear presentation of how the OHG system relates to human biology would strengthen the overall manuscript.
2. Several figures report data obtained using "n=2-3 gels or tissues". What is the rationale for use of such a small sample size? With regard to tissues, this draws into question the potential biologic relevance/similarity of the system under study.
3. Interpretation of Fig. 1 (lines 98-100) refers to similar values obtained relative to peritoneal connective tissues and peritoneal and omental adipose tissues. What is meant by 'peritoneal connective tissue' and from what specific site was it obtained? Similarly (as above) more detail is needed regarding 'peritoneal adipose tissue'. These comments are intended to clarify the potential biological relevance of the model under development. Again, it is unclear how many biologic subjects were examined (in extended data Fig. 1d, for example).
4. The spheroid data in Fig 1 and extended fig 2 also require additional rationale and information. What is the rationale for embedding spheroids? How were the spheroids formed and with how many cells? Were they all actually spheroids vs clusters or aggregates? What were the initial diameters? What is "relative spheroid area" relative to? Day 1 data need to be incorporated into extended fig. 2 for comparison.
5. The Methods section is uneven in that the bioengineering aspects of the study are described in detail while the biological methods are not. For example, in the "organotypic invasion assay" mentioned in Fig. 1, how was depth of invasion measured? How many cells were seeded? For how long? Were they individual cells or spheroids? Why or why not? Without this information, it is very difficult to determine the bioequivalence of the model.
6. It is unclear why invasion of OHGs is being compared to invasion of collagen gels, as these are two distinct aspects of the metastatic process. A better comparison would be comparison of OHGs to adipocytes.
7. The sentence (146-148) that the tropism shown by OC cells for adipose tissues is recapitulated in OHGs is overstated based on the data shown.
8. The rationale for the switch from spheroids in Fig. 1 to individual cells in Fig. 2 should be clearly stated in the manuscript.
9. The rationale for the comparative gene set enrichment analysis of differentially expressed genes between OHG-invading and non-invading cell lines (extended fig. 7) is unclear as these cell lines were derived from distinct patients with distinct mutational profiles etc. More useful would be a comparison of gene expression prior to vs during or after invasion to evaluate functionally relevant genes. The finding of TGF β regulating cell invasion is not novel.

10. Experiments with the NB-HA functionalized OHGs are of potential interest, but again, why such a small 'n' in the characterization of these reagents (n=2-3)?
11. Data shown in Fig. 4 regarding deformation of microdroplets or adipocytes at sites of OC cell contact require additional information and quantitation. How is 'deformation' defined? Controls where this deformation value is also measured on OC cell-free systems are needed. Moreover, the followon experiments shown in Fig. 4d switch to spheroids again. There is no reason to anticipate that cells will use the same migratory mechanisms when encountering an interface as an individual cell vs when migrating from a spheroid. Rationale for the experimental design should be incorporated into the manuscript.
12. Experiments in which the 'oil volume fraction' is altered (Fig. 5) are interesting. However in omental metastases, when adipocyte diameter is reduced and ECM deposition increases, this is accompanied by enhanced stiffness of the ECM that is not taken into account in these analyses. I.e., it is likely not just the decrease in adipocyte size that facilitate invasion. Also of interest would be to determine the MMP-dependence of invasion in this system.
13. Are these findings unique to ovarian cancer cells?
Minor points:
14. Please clarify what is meant on line 97 by "with identical collagen concentrations in the aqueous phase"
15. In figures that contain yellow arrowheads, it should be indicated in the figure legend what the arrowhead are attempting to designate.
16. References should be provided for the GFOGER motif in collagen (line 211).

Reviewer #2

(Remarks to the Author)

This manuscript present novel biomimetic organo-hydrogels (OGH) mimicking the microstructure, mechanics, and ECM of human adipose tissues. The hydrogels consist of oil microdroplets galeted with collagen I as the ECM phase to form the tissue-mimicking structure. These OHG's are used to investigate the invasion of ovarian cancer cells, as a function of systematically controlled OGH properties.

The OGH is thoroughly characterized, and their mechanical properties and structure are indeed similar to adipose tissue. The invasion experiments reveal several novel and interesting phenomena. Anisotropic mechanical properties and microstructure play an important role; the invasion is independent of matrix degradation; the invasion requires contractile cellular forces generating migratory tracks at the adipocyte-ECM interface. The manuscript is clear, and the used methodology is sound. Overall, I think the manuscript can be suitable for publication in Nature Communications, because of the novelty of the OGH and its use in cancer invasion investigations, and the interesting observations made regarding invasion.

Two comments that need to be addressed:

- The manuscript mentions that migration along the adipose-ECM interface, i.e. along the oil microdroplets in the model, is a key mechanism for invasion. However, the oil microdroplets do not present adhesive ligands with which the cells can interact. What, then, is the mechanisms of the interaction? This should be explained more clearly.
- Microdroplet size is found to influence invasion, and in fact the mode of migration. Interestingly, collective migration is found for bigger droplets (with larger inter-droplet space) and single cell migration is found for smaller droplets (with smaller inter-droplet space).
 - o How does this compare with migration modes depending on confinement as found by others? For example, other research has shown that even mesenchymal cells move collectively rather than individually along confined trails despite their labile cell-cell junctions (see, e.g., work of Friedl and coworkers, such as Haeger *Biochim. Biophys. Acta* 2014). This seems to be contradictory with the results of the manuscript.
 - o Then, TGF-beta treatment (inducing a mesenchymal phenotype) was found to have the largest effect on migration in the smallest droplet OHGs. However, in these smaller structures the behaviour was more mesenchymal in the first place (since migration was single-cell). Hence, one would expect a bigger influence of TGF-beta in the larger droplet OHGs. Can the authors reflect on this?

Version 1:

Reviewer comments:

Reviewer #1

(Remarks to the Author)

The authors have provided a detailed response to the critique and modified the manuscript to include additional details regarding methodology and experimental rationale. The sample numbers and replicates have been increased to reflect better rigor. Significant additional new data have been added.

Reviewer #2

(Remarks to the Author)

The authors have addressed all comments I raised in an appropriate way, and made corresponding changes to the manuscript. In my opinion, the revised manuscript is suitable for publication in Nature Comm.

Point-by-point Response

Reviewer #1 (Remarks to the Author):

This manuscript by Gonzalez-Molina and coworkers describes the generation and characterization of biomimetic organo-hydrogels to mimic the mechanical properties of visceral adipose tissues and characterize the behavior of ovarian cancer (OC) cells in these gels in comparison to adipose tissue in attempt to delineate the tropism of metastasizing OC cells to adipose. Mechanistic studies implicate a role for TGF β in this process. While the study is of interest, several points should be addressed. Major points:

1. The rationale for the design of the organo-hydrogels (OHGs) requires further information. It is shown in Fig. 1 that 70 μ m silicone oil microdroplets were used to mimic visceral adipocytes. Additional information is required on the system that is being mimicked. The methods just indicate 'macroscopically normal omental and peritoneal tissues' were examined. Presumably these tissues are from OC patients and are therefore unlikely to be normal and to not already contain OC cells (particularly the omental tissue). If peritoneal tissue, where, specifically, in the peritoneum were the tissues obtained? Were tissues stained for OC cells? Some of the data in Extended Fig. 1 would be better incorporated into Fig. 1, particularly data showing similar diameter. However this figure needs additional detail as well. It is indicated that n=170 cells were measured to determine diameter. From how many patients were these values obtained? Was patient age or overall adiposity (body mass index or similar) taken into account? A more clear presentation of how the OHG system relates to human biology would strengthen the overall manuscript.

Authors' response:

1. To analyse the "macroscopically normal" tissues, we have quantified PAX8+ cells in these tissues, showing very low presence, ($0.38 \pm 0.21\%$ PAX8+ nuclei (s.e.m.); 0.75 ± 0.44 PAX8+ nuclei mm⁻² tissue (s.e.m.)), as stated in the results text (lines 90-93). As "normal omental tissue" was not used throughout the study due to low availability, we have removed it from these analyses. We also note that a detailed analysis of the microstructure, mechanics and biochemical composition of normal and diseased tissues was previously reported by Balkwill and co-workers (DOI: 10.1158/2159-8290.CD-17-0284): adipocyte diameters reduced from 70 to 25 μ m, whilst the volume fraction of adipocytes reduced from 90 to 60%, with increasing disease scores. This range of microstructures is well captured by our models and cell invasion correlates well with these data. Further discussion of these observations and correlations is presented in lines 323-341.
2. We have modified the methods section to more precisely explain how the tissues were obtained (see "patient samples", lines 466-479).
3. We have moved the data regarding adipocyte diameter (New Fig. 1c) and included more measurements (N = 7 patients). We have also included quantification of the % volume fraction occupied by adipocytes and compared it to our OHG model (New Fig. 1d). This further shows that the tissues used in our study mainly comprise adipocytes (91% of the tissue volume).
4. Patient age and body mass index was not considered in selecting patient samples; however, we have included these parameters into a table (New Supplementary Table 1). This tissue heterogeneity allowed us to investigate, for

example, how adipocyte size affects cell invasion (New Fig. 5f,g, and Supplementary Fig. 14a).

2. Several figures report data obtained using “n=2-3 gels or tissues”. What is the rationale for use of such a small sample size? With regard to tissues, this draws into question the potential biologic relevance/similarity of the system under study.

Authors' response: to overcome the limited sample size, we have added more patient tissues and gel samples to all our analyses. The modified figures include: New figure 1c,d (N = 7 patients and 3 gels); New Fig. 1e,f (N = 5 gels, N = 5 (connective) and N = 9 (adipose) peritoneal tissues); Fig 3b (N = 3-5); Fig 5b (N = 4-5).

3. Interpretation of Fig. 1 (lines 98-100) refers to similar values obtained relative to peritoneal connective tissues and peritoneal and omental adipose tissues. What is meant by ‘peritoneal connective tissue’ and from what specific site was it obtained? Similarly (as above) more detail is needed regarding ‘peritoneal adipose tissue’. These comments are intended to clarify the potential biological relevance of the model under development. Again, it is unclear how many biologic subjects were examined (in Supplementary Fig. 1d, for example).

Authors' response:

1. Peritoneal connective tissue is part of the peritoneum that was extracted that did not present visible fat tissue. Peritoneal adipose tissues are parts of the peritoneum that present visible fat tissue. These tissues were selected by expert surgeons and validated by microscopy through staining for collagen (rich in connective tissue) and oil (rich in adipose tissue). We have modified the methods to better explain how these tissues were obtained (peritoneum with or without visible underlying fat; lines 472-475).
2. In Supplementary Fig. 1d (new Supplementary Fig. 1b), N = 3 patient tissues, this information has been added to the figure caption.

4. The spheroid data in Fig 1 and extended fig 2 also require additional rationale and information. What is the rationale for embedding spheroids? How were the spheroids formed and with how many cells? Were they all actually spheroids vs clusters or aggregates? What were the initial diameters? What is “relative spheroid area” relative to? Day 1 data need to be incorporated into extended fig. 2 for comparison.

Authors' response:

1. The embedded spheroid model was applied to investigate cell invasion following initial colonisation of the tissue. This is commonly used to study the impact of the surrounding matrix on (cancer) cell growth and invasion (e.g. Elosegui-Artola, Nature Materials, 2023). This has been clarified in our revised manuscript (lines 117-120).
2. We have added a section in methods “Cell invasion assays” (lines 515-531) to include details of how the experiment was performed (e.g. 3000 cells were seeded into u-bottom ultralow attachment plates for the forced aggregation of cells. Aggregates/spheroids were allowed to form over 4 days). We have also added details on what relative spheroid area refers to (i.e. total spheroid area /

average area of the control, for example spheroids embedded in collagen, or spheroids treated with DMSO for inhibitor experiments).

3. We have added a comparison between Day 0 spheroid area and Day 7 spheroid area (New Fig. 1g). These results clearly show that Tyk-nu, Tyk-nu.CPR, OVCAR8, and CAOV3 cells are the only cell lines that grow/invade in OHGs, while these other cell lines remained with a comparable size to Day 0.

5. The Methods section is uneven in that the bioengineering aspects of the study are described in detail while the biological methods are not. For example, in the “organotypic invasion assay” mentioned in Fig. 1, how was depth of invasion measured? How many cells were seeded? For how long? Were they individual cells or spheroids? Why or why not? Without this information, it is very difficult to determine the bioequivalence of the model.

Authors’ response: we have added this information in a new section for “Cell invasion assays” (lines 524-531). Briefly: for OVCAR8 10,000 individual cells were added, which rapidly adhere to the tissue/OHG surface. For patient ascites cells (100 ul without varying cell concentration), these were added as they were obtained from patients, mostly as cell spheroids. They were allowed to invade the tissue/OHG for 7 days. Other methods sections we have expanded include “Patient samples” (as mentioned above), “Collagen-based hydrogels” (to specify how collagen features were analysed; lines 595-598) and “Microdroplet and adipocyte shape analysis” (lines 683-688).

6. It is unclear why invasion of OHGs is being compared to invasion of collagen gels, as these are two distinct aspects of the metastatic process. A better comparison would be comparison of OHGs to adipocytes.

Authors’ response:

We did make comparisons between invasion in OHGs and adipocyte rich tissues (e.g. original Fig 1 g/h vs Fig 1i/j and data presented in subsequent figures). Further comparison was made with collagen gels were used for two reasons. Firstly, because collagen, as the main ECM component of adipose tissues, was also used as the ECM phase of the OHGs (except in OHGs in which we had engineered the gel phase too). Thus, to understand the impact of microdroplets and associated microstructure of the microenvironment on invasion, in comparison with fibrillar microenvironments, we used collagen gels as control. Secondly, peritoneal tissues devoid of adipocytes (mainly connective tissue) are collagen-rich. To investigate whether cancer cells preferentially invade adipose tissue, rather than neighbouring connective tissue in the peritoneum, invasion into peritoneal connective tissues and collagen gels was compared to invasion in OHG/adipose tissue. This rationale is more clearly stated in our revised manuscript (lines 102-105 and 144-152).

To better compare cell invasion responses between OHGs and adipose tissue, we have performed direct comparisons between peritoneal adipose tissue (N = 3 patients) and OHGs (N = 3). These experiments showed comparable cell invasion in both models (New Fig. 1m,n; lines 151-152).

7. The sentence (146-148) that the tropism shown by OC cells for adipose tissues is recapitulated in OHGs is overstated based on the data shown.

Authors' response: we have modified the sentence to specify “ex vivo adipose tissues” (lines 160-162). We believe that, based on the data provided in Figures 1 and extended Figures 2-5 (including in revised Fig. 1m,n), this sentence is not an overstatement.

8. The rationale for the switch from spheroids in Fig. 1 to individual cells in Fig. 2 should be clearly stated in the manuscript.

Authors' response: we switched to individual cells to better separate the impact of cell-cell adhesion from cell-matrix sensing. In addition, the observation of comparable (qualitatively at least) migratory phenotypes in both models is a useful further validation of the relevance of our model. We have modified the text to clarify the rationale (lines 167-170).

9. The rationale for the comparative gene set enrichment analysis of differentially expressed genes between OHG-invading and non-invading cell lines (extended fig. 7) is unclear as these cell lines were derived from distinct patients with distinct mutational profiles etc. More useful would be a comparison of gene expression prior to vs during or after invasion to evaluate functionally relevant genes. The finding of TGFb regulating cell invasion is not novel.

Authors' response: we performed the comparison between OHG-invading and non-invading cells to understand why these differences were observed (as some of the cell lines did not migrate, comparison of gene set enrichment before and after invasion was not possible). However, comparison of the gene set enrichment analysis for the different lines investigated provided clues about their differential migratory phenotypes, suggesting that a reduced cell-cell adhesion, typical of more mesenchymal cells, can facilitate this invasion (a correlation relatively classic in the field of metastasis). Although the role of TGFb on cell migration is not novel, its impact on the migratory phenotype of cancer cells in adipose tissue-mimicking OHGs is, indicating that some of the important molecular processes regulating migratory phenotypes in distinct microenvironments are shared, even though cell migration in microenvironments with a broad range of microstructure and mechanics can vary widely.

10. Experiments with the NB-HA functionalized OHGs are of potential interest, but again, why such a small 'n' in the characterization of these reagents (n=2-3)?

Authors' response: we have now added more replicates to our analysis (n=3-5).

11. Data shown in Fig. 4 regarding deformation of microdroplets or adipocytes at sites of OC cell contact require additional information and quantitation. How is 'deformation' defined? Controls where this deformation value is also measured on OC cell-free systems are needed. Moreover, the follow on experiments shown in Fig. 4d switch to spheroids again. There is no reason to anticipate that cells will use the same migratory mechanisms when encountering an interface as an individual cell vs when migrating from a spheroid. Rationale for the experimental design should be incorporated into the manuscript.

Authors' response:

1. As outlined qualitatively in the text, “the microdroplets and peritoneal adipocytes in contact with OVCAR8 cells presented deformed interfaces at contact points” (lines 261-263), which are visible alterations of the shape as shown, in Fig. 4b.
2. We have now quantified these deformations comparing droplets and adipocytes in contact with cancer cells (and not deformed by other droplets or adipocytes, which is commonly observed in crowded tissue/OHG areas; information added to methods section “microdroplet and adipocyte shape analysis” (lines 683-688)). This analysis shows significant changes in circularity (New Supplementary Fig. 9a).
3. Given that 1) the organotypic and spheroid models showed qualitatively comparable results, and 2) once cells have initiated migration within corresponding tissues they perceived comparable microenvironment in both scenarios (spheroid vs surface seeding), we performed most of our analyses using the spheroid model due to ease of imaging and quantification. Moreover, as explained in response to comment #4, we have modified the text to distinguish between the two models, which correspond to two different steps of tissue invasion.

12. Experiments in which the ‘oil volume fraction’ is altered (Fig. 5) are interesting. However in omental metastases, when adipocyte diameter is reduced and ECM deposition increases, this is accompanied by enhanced stiffness of the ECM that is not taken into account in these analyses. I.e., it is likely not just the decrease in adipocyte size that facilitate invasion. Also of interest would be to determine the MMP-dependence of invasion in this system.

Authors' response:

1. We acknowledge that migratory phenotypes in tumours could be regulated by multiple factors, from matrix microstructure and stiffness to biochemistry and associated molecular pathways. An increase in tissue stiffness observed in tumour samples recovered from patients with higher disease scores has indeed been reported (see Balkwill and co-workers DOI: 10.1158/2159-8290.CD-17-0284). However, we note that this could be expected not just from an increase in matrix density (tissues with higher disease score are associated with increase fibronectin and other ECM content), but also smaller adipocytes, even with equal mechanics in the gel phase, providing the volume fraction remains comparable. Our ability to dissociate these different parameters in OHGs is therefore particularly interesting. We note that the literature reported typically does not dissociate macroscopic mechanics from local nanoscale stiffness, even in the case of force probe microscopy, normally averaged and using relatively large probes (micron-scale colloids), in comparison to some of the very restricted gel phases associated with adipose tissues.
2. We have now revised the text of our manuscript to highlight that stiffness of adipose tissues increases in conjunction with reduced adipocyte size and increased total ECM volume fraction (lines 346-348). Although we do not investigate this effect in conjunction with reduced microdroplet/adipocyte size and increased ECM phase, our OHG model clearly demonstrates that cancer cell migration is modulated in adipose mimicking microenvironment with controlled microstructure (combined droplet size and volume fraction), independent of any

variation in local stiffness (although we stress again that macroscopic stiffness of corresponding materials would be expected to vary). Variation in local stiffness would be expected to further modulate (perhaps enhance) migratory phenotypes.

3. To further investigate the impact of adipo-mimetic OHGs on macroscale mechanics, in our revised manuscript we included additional rheology measurements of our models (New Fig. 5b,c and Supplementary Fig. 13c,d). These results show that while reducing microdroplet size enhances OHG stiffness, increasing the ECM volume fraction (while maintaining the local collagen concentration, and local mechanics) results in lower macroscopic stiffness. This therefore demonstrates an important role of tissue microstructure, independent of local mechanics, and that could synergise with local changes (increase) in matrix mechanics. This is further supported by the analysis of correlations between OVCAR8 invasion into peritoneal tissues and tissue stiffness or adipocyte size (only using tissues with low ECM volume fraction), showing that invasion correlates with adipocyte size strongly while it does not with stiffness (New Fig. 5f,g and Supplementary Fig. 13e).
4. We also show that invasion in 25 μm microdroplet OHGs with varying oil volume fraction is independent of MMP activity (New Supplementary Fig. 13b).

13. Are these findings unique to ovarian cancer cells?

Authors' response: these results are likely not unique to ovarian cancer cells while still relevant for other cancer types such as gastric, colorectal, breast, and leukaemia, as well as non-malignant cells (e.g. immune cells, stromal cells), which all show invasion/migration in adipose tissues. We have modified the discussion section to discuss this aspect and the importance of investigating these mechanisms in other cell types, which we are currently investigating in separate studies (lines 452-455 and 461-464).

Minor points:

14. Please clarify what is meant on line 97 by “with identical collagen concentrations in the aqueous phase”

Authors' response: we have modified this sentence to “with identical collagen concentration than in the aqueous phase of OHGs” (line 105) and further detailed this feature in our revised methods (“Collagen-based hydrogels”).

15. In figures that contain yellow arrowheads, it should be indicated in the figure legend what the arrowhead are attempting to designate.

Authors' response: we have added this information in all figure captions.

16. References should be provided for the GFOGER motif in collagen (line 211).

Authors' response: we have now provided a reference.

Reviewer #2 (Remarks to the Author):

This manuscript presents novel biomimetic organo-hydrogels (OGH) mimicking the microstructure, mechanics, and ECM of human adipose tissues. The hydrogels consist of oil microdroplets grafted with collagen I as the ECM phase to form the tissue-mimicking structure. These OGHs are used to investigate the invasion of ovarian cancer cells, as a function of systematically controlled OGH properties.

The OGH is thoroughly characterized, and their mechanical properties and structure are indeed similar to adipose tissue. The invasion experiments reveal several novel and interesting phenomena. Anisotropic mechanical properties and microstructure play an important role; the invasion is independent of matrix degradation; the invasion requires contractile cellular forces generating migratory tracks at the adipocyte-ECM interface. The manuscript is clear, and the used methodology is sound. Overall, I think the manuscript can be suitable for publication in Nature Communications, because of the novelty of the OGH and its use in cancer invasion investigations, and the interesting observations made regarding invasion.

Two comments that need to be addressed:

- The manuscript mentions that migration along the adipose-ECM interface, i.e. along the oil microdroplets in the model, is a key mechanism for invasion. However, the oil microdroplets do not present adhesive ligands with which the cells can interact. What, then, is the mechanism of the interaction? This should be explained more clearly.

Authors' response: The reviewer is correct in pointing out that not all of the systems present displayed ligands at the oil-gel interface. Indeed, in BSA-stabilised droplets dispersed in collagen gel, there may not be specific cell adhesive ligand at the interface, although collagen itself may assemble at the liquid interface (see collagen stainings in Extended Figures 1 and 3). However, in other engineered OGHs presented, using NB-BSA, integrin ligands are presented specifically at the interface and interfacial mechanics is found to modulate cell migration, highlighting the role of nanoscale mechanics in migratory phenotype (see Figure 4G-K). Similarly, blocking integrin $\beta 1$ practically abolished cell invasion into OGHs (Fig. 2e). We have modified our manuscript (and added representative images of spheroids in distinct inhibitor conditions (New Supplementary Fig. 6c) to further emphasize the dependence of the migratory phenotypes observed on ECM adhesion at droplet-gel interfaces (lines 181-187).

- Microdroplet size is found to influence invasion, and in fact the mode of migration. Interestingly, collective migration is found for bigger droplets (with larger inter-droplet space) and single cell migration is found for smaller droplets (with smaller inter-droplet space).

o How does this compare with migration modes depending on confinement as found by others? For example, other research has shown that even mesenchymal cells

move collectively rather than individually along confined trails despite their labile cell-cell junctions (see, e.g., work of Friedl and coworkers, such as Haeger *Biochim. Biophys. Acta* 2014). This seems to be contradictory with the results of the manuscript.

Authors' response: This is an interesting point, calling for further investigations. We have added a comparison of our model with the collagen density system from Friedl and co-workers (see text lines 367-374, associated with Fig 5h,i). We describe that, in 25 μm microdroplets, we have two aspects that influence cell migration, 1) enhanced physical barriers (which limits multiple cells migrating simultaneously, but could still allow collective migration of single cell strands), and 2) increased migration paths (more cells are in contact with the droplet-ECM interface, providing more migration paths). The second effect (more migration paths) may share similarities with low collagen gels at low density (increasing pore size, facilitating single cell invasion of mesenchymal cells as shown by Friedl and co-workers).

Moreover, we have added a comparison with invasion into peritoneal adipose tissues (New Supplementary Fig. 14a), showing a similar behaviour in large adipocyte tissues (collective migration), and small adipocyte tissues (single cell migration).

o Then, TGF-beta treatment (inducing a mesenchymal phenotype) was found to have the largest effect on migration in the smallest droplet OHGs. However, in these smaller structures the behaviour was more mesenchymal in the first place (since migration was single-cell). Hence, one would expect a bigger influence of TGF-beta in the larger droplet OHGs. Can the authors reflect on this?

Authors' response: we have modified the paragraph (lines 402-412) and added new analyses to emphasize that we are investigating whether a mesenchymal phenotype facilitates invasion in the more spatially restrictive 25 μm microdroplet OHGs. To do so, we use CAOV3 cells, which only show collective migration and present significantly lower invasion in 25 compared with 70 μm droplet OHGs (New Supplementary Fig. 17a,b). We further show in the revised manuscript that while TGF- β treatment enhanced invasion in both 70 and 25 μm droplet OHGs (New Supplementary Fig 17c), the effect of TGF- β is stronger on 25 μm droplet OHGs. This shows that smaller droplet OHGs not only can induce single cell migration (in specific cell lines), but also that, in these systems, single cell migration regulates invasion into these restrictive environments.